# Acceptance of AI-based tools in consumer financial decision-making: An application of the extended technology acceptance model

Tomasz Szopiński[1], Michał Buszko [2]*, Małgorzata Porada-Rochoń[3]

1 School of Business, VIZJA University, Warsaw, Poland, 2 Department of Financial Management, Institute of Economics and Finance, Faculty of Economic Sciences and Management, Nicolaus Copernicus University, Toruń, Poland, 3 Institute of Economics and Finance, Faculty of Economics, Finance and Management, University of Szczecin, Szczecin, Poland

☉ These authors contributed equally to this work.
* mibus@umk.pl

## Abstract

The development of artificial intelligence has led to an increase in scientific papers on its applications in financial services. Some of them focus on consumers, particularly presenting the aspects of automated servicing through robo-advisory systems. Nonetheless, relatively rarely does the research relate to the aspects of supporting consumers with AI-driven solutions for their own financial decision-making, which we find as a research gap. Our investigation focuses on finding how and to what extent AI-based tools empower consumers, providing support to their personal financial decisions. Our goal is to identify the determinants of acceptance of AI-based tools supporting financial decision-making, including perceptions of ease of use, usefulness, attitudes, and intentions to use. We base our work on the TAM model, for which we developed a dedicated scale to measure the individual items. The investigation is a pilot study that tests the survey on a representative sample of society. We gathered data through the CAWI survey research on a group of 371 respondents from three Polish universities. Our research shows that AI-based tools supporting financial decisions are primarily perceived through their usefulness and the potential to improve financial literacy. Furthermore, the PLS-SEM modelling confirms positive relationships between perceived ease of use (PEOU), perceived usefulness (PU), attitudes (ATT), behavioral intentions (BI), and actual use (AU) of AI-based tools in making financial decisions by consumers, except for the impact of PU on BI. We observed the strongest relation between PU and ATT towards AI-based tools, and between BI and AU. Our findings demonstrate that PU influences BI only indirectly through the construct of attitude (ATT). Such a phenomenon of a fully mediated relationship deviates from the traditional TAM assumptions. Our study confirms the coherence of the survey and the validity of the extended theoretical model of the acceptance and use of AI-based tools in consumers' financial decision-making.

**Data availability statement:** All data files are available from the ICPSR database (ICPSR-232047), https://www.openicpsr.org/openicpsr/project/232042/version/V1/viewhttps://doi.org/10.3886/E232042V1.

**Funding:** MB 41/2024/Debiuty_7 IDUB UMK https://idub.umk.pl/konkursy-i-wyniki/konkursy-dla-pracownikow/debiuty/wyniki-i-za-konczone-edycje-debiuty/edycja-vii-04-03-2024-r-05-04-2024-r/ The founder doesn't play any role in the study design, data collection and analysis, decision to publish, or preparation of the manuscript.

**Competing interests:** NO authors have competing interests.

## Introduction

AI is an intelligent learning system that utilizes mathematical algorithms to perform tasks traditionally associated with human cognitive features. It encompasses a broad spectrum of capabilities, including the ability to imitate human intelligence, operate with foresight, and autonomously analyze environments to take actions aimed at achieving specific goals. Such attributes make AI particularly suited for finance, where adaptive learning, behavioral analysis, and real-time responsiveness are critical for effective planning, decision-making, and control. One of the domains that can be revolutionized by AI is undoubtedly personal finance. In particular, AI-based solutions and apps can support consumers in the financial decision-making process, particularly in light of the dynamic changes in the financial market, the increasing complexity of financial products, and the heightened risk of financial and cyber fraud.

The growing adoption of AI in finance in recent years has rendered its application by financial institutions no longer optional, but necessary to meet customer expectations [1]. It has also triggered intensive scientific research in this field. The literature on AI in finance generally presents two major perspectives on AI's use: financial institutions (primarily banks) and customers [2]. The dual character of the research can be justified by the fact that AI strongly influences both the financial institutions' economic environment as well as customers' behavior. In both areas, the implementation of AI is considered to bring radical changes and require public policies [3]. The works devoted to financial institutions, like banks, conceptualize or demonstrate the AI impact on the strategies, processes, and customers [4], supporting the transformation [5], processing large amounts of structured and unstructured data, predicting market trends, gaining insights and identifying investment opportunities, which ultimately lead to better decision-making [6]. AI has also been found important for credit scoring [7–8], asset management [9], or marketing activities [10]. AI-based solutions are also considered suitable to solve the problems of bank performance, banking efficiency, or risk management [11]. The works in the second field, i.e., devoted to customers' behavior toward AI, include primarily the research on consumers' acceptance of AI in financial services. They present the consumer's characteristics supporting the use of financial services based on AI [12–13], acceptance of payments authorized by AI-based tools [14], or determinants of adoption of specific AI-based solutions, such as voicebots [15] or chatbots [16–19]. A visible part of the research has also become the papers related to the use of AI-based automated, consumer-adjusted robo-advice services [20–24], which are algorithmic, non-emotion-based, multiple-purpose savings and investment financial solutions. Some research on AI presents a more general approach to the acceptance of AI by consumers in banking [2,25], describing the reasons for AI adoption and the factors that deter consumers from adopting AI.

Despite growing research in the field of consumers' use of AI-driven financial services, we identify a new sub-area, which is so far relatively rarely explored, i.e., the acceptance of AI-based tools supporting autonomous personal financial decision-making. Due to the dynamic changes of the financial market, more complex financial products, as well as pressure on instant financial decisions, consumers, especially those with limited financial literacy, are particularly vulnerable

to misunderstanding or wrong assessment of terms, contractual conditions, pricing, or risk impact of financial services. AI-driven solutions provided by the banks, non-banking financial institutions (FinTechs), or other innovative units may effectively help in solving this issue by supplying fair, customer-adjusted information, stimulating financial education, as well as building financial skills and competencies.

Among AI-driven solutions (tools) supporting consumers, one can find primarily algorithms or applications providing financial data and information, generating prompts to guide users in performing financial tasks, issuing fraud alerts, delivering financial recommendations, comparing offers, and providing other forms of advice, such as automated support in solving problems. AI may then be used to explain financial terms, offer training in financial games, control the personal budget, plan the savings and investments, simulate loan repayment scenarios, prompt negotiation scenarios, make complaints with financial institutions [26], predict stock price movements [27] or recommend stocks to buy [28]. Such solutions, in contrast to robo-advisory, enable personal financial decision-making without automatic or semi-automatic AI-driven management. They include, e.g., Finchat, You Need a Budget, or TurboTax. Additionally, the consumer in the financial services market may be supported by AI-based tools, which are not directly dedicated to personal finance but can assist in making decisions. Here, one can mention general conversational model applications that offer universal advisory in various contexts through answering the questions, providing information and data, comparing products, or prompting solutions to specific problems. They include ChatGPT, Gemini, Bing Chat, Claude, Copilot, and others. Apart from generative AI, consumers' financial decisions can be supported by AI-based non-conversational software aimed at cybersecurity, which can effectively protect consumers against fraud.

The field of AI-based tools that support customers' financial decision-making seems to be particularly important, as the functionality of AI gives new possibilities and operating capacity to consumers. In general, consumers are the weaker party in relations with financial institutions. They are prone to take non-optimal decisions about their finances, buy expensive products, or products of elevated risk. Frequently, they are also less protected against fraud. Objectively, they might be the most beneficial group that takes advantage of AI-based support in finance. Nonetheless, because AI-driven solutions in finance are still quite new and not all their risks are identified or predictable at present, they can be perceived to some extent as controversial. It is not obvious that consumers accept them in all aspects of financial support. Just in case of ChatGPT, recent findings show that the application can assist in managing personal finance by providing general recommendations; however, its advice may still lack depth and personalization, highlighting both the potential and limitations of a generative AI agent as a financial assistant [29]. These ambiguous results underline the importance of empirical investigation on AI-based tools used in personal finance decision-making.

The research problem, which we identify here, is the assessment of the perception, attitudes, acceptance, and actual use of the AI-driven solutions that can be used to support autonomous financial decisions of the consumers, as well as an identification of the relationships between their constructs.

Despite the rising number of articles devoted to various aspects of AI in finance, so far, the research has not been focused on the aspects of supporting independent financial decision-making driven by AI-based tools, which is the core of our investigation. The work, which in some part covered the issue and was an inspiration for our research, was [30], where the authors analyzed the impact of socioeconomic factors on the use of modern technologies in the process of personal finance management. Focusing on robo-advisory as well as a supporting solution, they discovered the preference for using computer programs to monitor spending habits rather than automated investment decisions. The higher significance of the former gave us a base to commence own research on AI-based tools that support consumers' financial decisions and to open a discussion on how AI support in personal financial decision-making is perceived, what attitudes towards it are, why, and to what extent it is accepted and used, giving the field to explore in depth. The stimulus for conducting the research in the mentioned scope is also the fact that attitudes toward AI are multidimensional and influenced by a complex interplay of social, technological, and psychological factors, often polarized, distinguishing them from attitudes toward earlier technologies [31].

The goal of our paper is to identify the determinants of the acceptance and the actual use of AI-based tools by consumers in their personal financial decisions, including perception of the ease-of-use, usefulness, attitudes, and behavioral intentions. Furthermore, we intend to find relations between the determinants as well as areas of consumers' financial decision-making with the strongest impact of AI-based tools.

We design our research on the framework of the Technology Acceptance Model (TAM) proposed by Davis [32], which provides a theoretical basis for the assessment of five areas related to AI acceptance by consumers. The use of TAM and its modified versions can be pointed out as the most common theoretical base for AI investigation [33], and hence, it will be a base for our research. The mentioned model will be used as a platform to address the research problem of the application of AI-based tools to support the process of financial decision-making. We call the Technology Acceptance Model adjusted to the specifics of our research the extended TAM.

In our work, first, we conducted extensive literature studies. Due to relatively limited research in scope of the use of AI-based tools for supporting independent financial decisions of consumers, we studied the works related to attitudes of decision-makers to the use of AI in banking activities, the attitudes of Internet users to AI-driven mobile applications related to finance, the attitudes to robo-advisory services, which cover the aspects closely related to our core investigation. Apart from that, we conducted a research survey from February to October 2024 among a group of respondents from three universities in Poland. Poland has a modern, digitally mature banking system, characterized by a common acceptance of innovative banking solutions and FinTechs, especially among young people [19]. Consumers here are open to innovation, creating demand for intelligent and personalized solutions [34]. That provided us with a solid foundation for investigating AI-based solutions. The survey was used as a pilot investigation for the full research project planned on a representative sample of the society. Totally 371 participants took part in our survey. To measure attitudes toward artificial intelligence, we applied the General Attitudes Towards Artificial Intelligence Scale (GAAIS) developed by Schepman and Rodway [35], adapted to the context of our research. The remaining constructs — perceived usefulness of AI solutions, ease of use, intentions regarding AI use, and actual AI use — were measured using our own scale, developed specifically for this study. The research questionnaire was structured into five spheres: perceived usefulness of AI-based tools, ease of use, attitudes towards AI, behavioral intentions, and actual use of AI-based tools. The survey results were modelled using PLS-SEM, supported by SmartPLS 4 software.

Our paper is structured as follows: first, we present the theoretical background for our research; then, we describe the hypotheses and methods; next, we present the results; and finally, we conduct a discussion and draw conclusions.

## Theoretical background and literature review

Artificial intelligence originates from advancements in computer science and engineering. Despite being based on technology, it requires human-initiated engagement, capabilities, competencies, and effort, all of which are enhanced by purposeful innovation [36]. Such conditions clearly differentiate AI from human intelligence determined by natural selection and neurological background [36–37]. From a broader perspective, AI can be seen as a system's ability to correctly interpret external data, learn from it, and utilize the learning to achieve specific goals and tasks through flexible adaptation [38–39]. It is the ability of machines, e.g., computers or computer-controlled robots, to perform tasks commonly associated with intelligent beings and exhibit human characteristics, i.e., to conduct reasoning processes and plan or take specific actions in identified environmental conditions [40]. AI can be considered, then, as a philosophy of making machines operate, think, behave, and perform either the same or like humans [41]. The presented features make AI particularly useful in financial management, supporting the weaker party in financial transactions, i.e., consumers. The ability to process large amounts of data, adapt to a constantly changing economic environment, and operate without emotion in financial markets makes AI-based tools particularly suitable for providing effective, automated, informative, or computational services to consumers, supporting their autonomous financial decision-making.

Research in the field of AI-driven tools for the financial services market, which support decision-making processes, is still relatively limited. Works on the acceptance of AI in the consumers' financial decision-making primarily relate to the use

of language models (e.g., ChatGPT). Recent findings reveal a substantial improvement in financial reasoning capabilities between GPT-3.5 and GPT-4, with the latter demonstrating near-perfect performance in financial literacy tasks. This rapid improvement highlights the growing potential of generative AI to serve as a reliable source of financial advice for consumers [42]. The work of [27] demonstrates that language models can be utilized to predict stock market returns using news headlines. Similarly, recent research demonstrates that ChatGPT can extract key insights from financial news articles, helping to reflect overall market performance and support the development of effective investment strategies [43]. Findings from [44] suggest that ChatGPT offers superior financial advice for one-time investments compared to robo-advisors. It can also correct the optimistic biases of human analysts [45]. Analyses performed by [29] show that the effective use of language models in financial decision-making depends on the task's goal, and the choice between models depends on the priorities of the user, such as accuracy, cost, or the quality of textual output. Recent findings also indicate that generative AI chatbots, including ChatGPT, Bing, and Bard, can provide investment recommendations that are generally rated as medium to high in terms of relevance, accuracy, specificity, and justification. Their performance was evaluated across different investment amounts and market contexts, suggesting consistency and broad applicability of such tools in supporting individual investment decisions [46]. Moreover, ChatGPT can be used to generate investment portfolios adapted to different risk preferences, supporting its applicability in personal financial management [47]. Additionally, ChatGPT has been shown to assist in asset selection and portfolio diversification, with its recommendations leading to portfolios that outperform randomly constructed ones and exhibit greater diversification, confirming its potential as a tool for building efficient investment strategies [48].

Research conducted by [49] indicates that AI holds considerable potential in finance and accounting education. Among other things, it can make recommendations to households on managing their own finances. Research also shows that AI plays an increasingly empowering role in enhancing financial decision-making, fostering greater user confidence and financial capability. For example, those who use financial apps for mobile phones exhibit greater capability and confidence in improving their financial decision-making through exposure to technology [50]. Furthermore, recent studies have begun to explore the integration of generative AI models such as ChatGPT into robo-advisory systems, emphasizing their potential in portfolio optimization, market sentiment analysis, and decision support [51]. These applications confirm the growing relevance of AI tools in consumer finance and further justify the need to investigate user acceptance of such technologies.

Based on the literature and theoretical framework proposed by other authors, we designed our research to find the relations between the ease of use, perceived usefulness, attitudes towards AI, behavioral intentions, and actual use of AI tools in consumer finance management to support consumers ' decision-making. We then applied the TAM model by Fred Davis [32], which we named the extended TAM, as the model was adjusted to our specific needs. The literature review and TAM model supported us in the formulation of hypotheses, which we put into the verification during our research process. The original Technology Acceptance Model (TAM) is a framework designed to explain how users accept and use a new technology. It focuses on two key factors that influence the adoption of technology: perceived usefulness and perceived ease of use. These two factors directly influence attitude toward technology and behavioral intention to use it. Ultimately, behavioral intention to use determines actual use. Perceived ease of use refers to "the degree to which a person believes that using a particular system would be free of effort" [32]. In turn, perceived usefulness is defined as „the degree to which a person believes that using a particular system would enhance his or her job performance" [32]. Attitude towards technology refers to individuals' perceptions, emotions, and beliefs about various technologies. These attitudes encompass both emotional aspects (e.g., positive or negative feelings) and cognitive dimensions (e.g., perceived functionality, usefulness, or societal importance). Attitudes toward AI are more multidimensional and influenced by a complex interplay of different factors, making them distinct from attitudes toward earlier technologies [31]. Behavioral intentions are indications of a person's readiness to perform a behavior. This readiness to act can be operationalized by asking if people who intend to engage in the behavior are truly willing to do this [52].

Based on the TAM assumptions (which in the scope of AI we labeled as the extended TAM), we expect that the perceived ease of use of AI-based tools in making financial decisions will positively influence the usefulness of AI and attitude towards AI. Perceived ease of use and perceived usefulness are the two most essential elements in accepting technology systems, specified by Davis [32]. In the scope of new technologies in finance, [53] found that perceived ease of use of mobile banking has a positive effect on the perceived usefulness of mobile banking and on attitude towards mobile banking. Moreover, the perceived ease of use of an electronic wallet positively affects perceived usefulness and intention to use such a solution [54]. In addition, studies on digital payment platforms show that perceived ease of use positively affects perceived usefulness in the context of consumer adoption [55]. For AI-based tools [20–21] stated that the perceived ease of use of financial robo-advisors has a positive effect on their perceived usefulness. Also, perceived ease of use has been shown to influence attitudes toward adopting new technologies. For example, research on consumer acceptance of the Bio-QR code traceability system demonstrated a significant positive relationship between perceived ease of use and attitude toward adopting this technology [56]. Similarly, the perceived ease of use of financial robo-advisors has a positive effect on attitudes toward them [20–21]. Furthermore, research on the acceptance of artificial intelligence in general confirms that perceived ease of use significantly influences both perceived usefulness and attitudes toward AI technologies [51,57]. Continuing this line of reasoning, previous research on the acceptance of artificial intelligence in mobile banking also confirms that perceived ease of use significantly influences both perceived usefulness and attitude toward the technology [58]. Along with the presented research, we propose the following hypotheses devoted to the impact of perceived ease of use of AI tools in financial decision-making on perceived usefulness and attitude towards it:

$H_1$: Perceived ease of use positively affects the perceived usefulness of AI tools in consumer financial decision-making.

$H_2$: Perceived ease of use positively affects attitude towards AI tools in consumer financial decision-making.

Next, previous research on the acceptance of artificial intelligence technologies confirms that perceived usefulness significantly influences attitudes toward AI solutions [57]. A paper by [22] shows that the perceived usefulness of robo-advisory services positively influences attitudes towards the technology. [20] reached the same conclusion. In addition, research on the adoption of artificial intelligence in mobile banking confirms that perceived usefulness significantly affects attitude toward the technology [58]. Moreover, studies on other innovative technologies, such as the Bio-QR code traceability system, also show that perceived usefulness has a significant positive impact on users' attitudes toward adopting the solution [56]. These results are consistent with our expectation that perceived usefulness plays an important role in shaping user attitudes toward AI tools in financial decision-making.

In the context of artificial intelligence applied to mobile banking, [58] reported a significant effect of perceived usefulness on the intention to use such technologies. Similarly, for other types of digital financial tools, [59] found a positive relationship between the perceived usefulness of QR code mobile payment systems and behavioral intention to use them. [54] also showed that the perceived usefulness of an electronic wallet positively influences the intention to use it. Additionally, research on the acceptance of digital payment platforms confirms that perceived usefulness is a key determinant of users' behavioral intention to adopt such tools [55]. Based on these works, we proposed the following hypotheses regarding the use of AI tools in financial decision-making:

$H_3$: Perceived usefulness positively affects the attitude toward AI tools in consumer financial decision-making.

$H_4$: Perceived usefulness positively affects the behavioral intention to use AI tools in consumer financial decision-making.

Thirdly, we expect that positive attitudes toward AI imply the behavioral intention to use AI-based tools in support of financial decision-making as well as their actual use. Research on technology acceptance in the context of artificial intelligence confirms that attitude toward AI significantly influences the behavioral intention to use such technologies [57]. Our expectation is backed up by the results of [60], who stated that users' attitudes towards generative AI positively influenced their intentions to use these technologies. A similar relationship was observed by [61] when analyzing attitudes to robo-advising. The work proves that the user's attitudes towards AI-based advisory significantly impact adoption, actual use,

and perceived benefits of AI. [62] found a positive relation between attitude towards AI in the banking industry and behavioral intention of using it. [53] stated a positive relationship between attitude towards mobile banking and the intention to use it. In addition, research on the adoption of FinTech services confirms that attitude towards such technologies significantly influences users' behavioral intentions to adopt them [63]. [64] analyzed the acceptance of the financial mobile applications. Their paper shows a positive relationship between attitude towards mobile financial apps and intention to use them. Furthermore, both the original Technology Acceptance Model by Davis [32], which posits that behavioral intention directly influences actual technology use, and more recent empirical research confirm that behavioral intention has a significant positive effect on actual usage behavior—for example, in the context of mobile banking [65]. We propose the following hypotheses:

$H_5$: Attitude towards AI tools in consumer financial decision-making has a positive effect on behavioral intention of using them.

$H_6$: Behavioral intention of using AI tools in consumer financial decision-making affects their actual use.

Based on the concept of the TAM model, we expect that the relationships between constructs in the acceptance of AI tools may be mediated by latent variables, leading to both direct and indirect effects. For example, prior research on robo-advisory services has shown that perceived usefulness mediates the relationship between perceived ease of use and attitudes toward the technology [20]. Other studies have reported mediation effects in the relationship between perceived usefulness and behavioral intention. [64] observed that attitude toward mobile financial applications mediates the link between perceived usefulness and the intention to use such apps. Similar observations were made by [13] in the context of mobile banking, and comparable mediating patterns were also reported in studies examining the application of AI technologies [66]. Additionally, [67] confirmed that attitude mediates the relationship between perceived usefulness and behavioral intention in the context of mobile banking. Although these findings concern mobile financial applications, AI-based tools can be considered a more advanced form of digital financial technology. This justifies transferring the conceptual assumptions between these domains. In support of this, [68] also reported that attitudes toward AI mediated the link between perceived usefulness and intention to adopt AI in banking operations. Therefore, we may formulate the following hypotheses:

$H_7$: The relationship between perceived ease of use and the attitude toward AI tools in consumer financial decision-making is mediated by perceived usefulness.

$H_8$: The relationship between perceived usefulness and the behavioral intention to use AI tools in consumer financial decision-making is mediated by the attitude towards AI tools.

## Materials and methods

The empirical component of our work is designed as a pilot study for the comprehensive research project on the use of AI-based tools in supporting financial decision-making by consumers throughout society in Poland. We performed convenience sampling using the CAWI technique on a group of students from three universities in Poland (in Toruń, Szczecin and Warsaw) and their relatives. Access to the research questionnaire was through the QR code posted on the websites of the university faculties participating in the survey. The data collection period was from February to October 2024. Totally 371 participants took part in our survey. Respondents could only submit the data if the survey was completed in full. Due to the distribution of the questionnaire via the websites, we could not determine the precise response rate.

In general, participants of the survey represent a group of young people with relatively high economic literacy and skilled in digital solutions, which may limit the generalizability of the results. The influence of age on financial behavior and decisions is recognized as one of the key research areas in modern finance, and especially young customers are more willing to use innovative solutions, e.g., FinTechs [19]. Although Poland went through the economic and political transition starting from 1989, we do not address this issue in our current research, as the respondents of our survey are primarily young people born after the changes. They were then raised in a digital culture and grew up in similar conditions to their

peers in Western countries [19]. The research in general confirms the higher inclination, better perception of the potential, or higher level of acceptance of using modern technologies in finance, including AI, by younger generations than older people [31,69,70]. However, the relationship can be nuanced and not very straightforward, as different factors of AI adoption may matter for different generations [71]. In the work of [31], presenting an approach to both robo-advisory and AI-based support in various countries, younger people are more willing to use both solutions than older people. In contrast, the impact of the level of education is ambiguous and does not show a clear pattern among the countries. The impact of age on the acceptance of robo-advisory and AI-based support just for Poland turned out to be statistically insignificant in the first case and significant and negative in the latter [31]. The impact of education was not statistically significant. Based on such findings, we may expect that results from our sample may show stronger or more visible relations between the constructs of the model than the representative sample of the Polish society. The group of our respondents mainly consists of young people, who are in general more inclined to risk, innovations, new technologies, and are also more digitally active than older generations.

The intention of the sampling we used was to verify the validity of the questions and constructs, as well as the consistency of the research questionnaire, before starting the full-scale survey on the representative sample.

The period of data collection has resulted from the organization of the semesters and a summer break during the academic year in Poland. In the collecting period, we did not identify any fundamental events that could have a particular impact on AI use in Poland (e.g., changes in law, institutional changes, problems of AI misuse, public AI campaigns, and starting the dedicated courses on AI at the universities taking part in the research), and could clearly affect the respondents. Moreover, as the AI-based tools in finance are rather new, we expect the potential changes in the adoption will appear rather in the medium-to-long term when the experience in their use increase. We are also not able to address potential temporal effects, as so far in Poland, there is no periodic research on the attitudes toward AI in finance.

In the survey research, we use our own questionnaire elaborated on an adjusted Schepman and Rodway scale model, i.e., the General Attitudes Towards Artificial Intelligence Scale (GAAIS), adapted to the context of our research. To measure consumer attitudes toward the use of AI-based tools in their financial decision-making, we adapted the General Attitudes Towards Artificial Intelligence Scale (GAAIS) developed by Schepman and Rodway in 2022. While the original version of the scale assessed general attitudes toward AI in everyday life, our study aimed to capture a more specific and functional perspective—how consumers evaluate AI applications in the domain of personal finance.

The adaptation process consisted of several stages. First, we conducted a content analysis of the original GAAIS items, identifying those that could be logically reformulated to fit the financial context. General items referring to the functions, emotions, and consequences of AI were reworded to explicitly reflect situations in which consumers use AI tools in managing their own finances. For example, the original statement "Artificial intelligence can have positive impacts on people's well-being" was adapted to "The use of artificial intelligence in consumers' financial management can have a positive impact on their financial well-being." This type of linguistic adaptation aimed to preserve the original meaning while grounding each item in financial realities.

Simultaneously, we excluded those original items that did not fit the study's context – particularly those referring to macroeconomic applications, professional environments, or highly general and unspecified aspects of daily life. In their place, we added new statements addressing contemporary consumer dilemmas related to the use of AI in finance, such as the possibility of algorithmic errors, the risk of losing control over financial decisions, and concerns related to privacy and surveillance.

A crucial stage of the process was the linguistic adaptation of the scale into Polish, which was conducted in accordance with the back-translation procedure. Initially, two independent translators with expertise in psychological and technological terminology translated the items from English into Polish. Subsequently, another translator, unaware of the original, retranslated the Polish version back into English. The two English versions were compared, and any discrepancies were

analyzed and refined to ensure semantic and contextual consistency. Special attention was also paid to ensuring that the final Polish wording was clear, unambiguous, and aligned with financial language used in real-life contexts.

As a result, the final version of the scale consists of 16 unique statements. The scale is balanced in terms of emotional tone (positive and negative attitudes) and covers all three components of attitude: cognitive (assessments of AI's effectiveness, safety, and usefulness), emotional (enthusiasm, anxiety, concern), and behavioral (preferences for using AI instead of humans).

The questions in the research questionnaire were structured according to the TAM model and divided into spheres: perceived usefulness of AI solutions, ease of use of AI solutions, attitudes towards AI solutions, intentions regarding the use of AI in the future and actual use of AI. Most participants were aged between 18 and 24 years, with fewer individuals represented in other age groups, including those under 18, aged 25–29, and 30 and above. In terms of gender, women constituted most of the sample. The sample characteristics are presented in Table 1

The numerical part of the research was based on PLS-SEM modelling and performed with the support of SmartPLS 4 software.

Fig 1 presents the concept and directions of our SEM modelling with 5 latent variables.

Table 2 shows the loadings describing the surveyed latent variables and the corresponding descriptive statistics. We totally used 40 variables (questions) divided into 6 constructs. We used the Likert scale (1–5) for the assessment of each variable. Most of the variables present a mean in the range of 2.5–3.5, suggesting that respondents' answers fluctuated mainly in the middle of the scale. The highest mean was recorded for the PU2 variable (3.91) and PU3 variable (3,80), suggesting that this variable was rated the best by respondents. In contrast, the lowest averages occurred

**Table 1. Sample characteristics.**

| age | Number | % |
|---|---|---|
| under 18 | 12 | 3,23 |
| 18-24 | 331 | 89,22 |
| 25-29 | 18 | 4,85 |
| 30 and above | 10 | 2,70 |
| gender | Number | % |
| men | 127 | 34.2 |
| women | 244 | 65.8 |

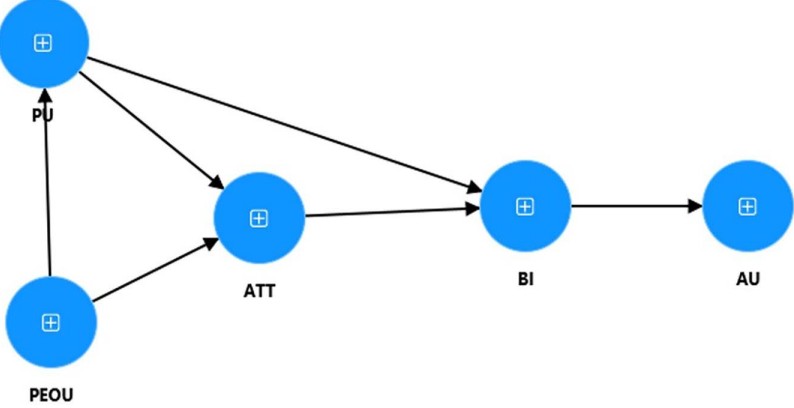

**Fig 1. Conceptual model.**

**Table 2. Descriptive statistics.**

| Items | Mean | Median | Moderate | Std. | Min. | Max. |
|---|---|---|---|---|---|---|
| **Perceived usefulness** | | | | | | |
| PU1. The use of AI-based solutions makes consumers' financial needs better met | 3.36 | 3.00 | 4 | 0.943 | 1 | 5 |
| PU2. Thanks to solutions based on artificial intelligence, one can gain faster knowledge about finance | 3.80 | 4.00 | 4 | 0.995 | 1 | 5 |
| PU3. With AI-based solutions, one can get quickly answers to the financial questions | 3.91 | 4.00 | 4 | 1.015 | 1 | 5 |
| PU4. AI-based solutions can find profitable investments in the financial market | 3.25 | 3.00 | 3 | 0.998 | 1 | 5 |
| PU5. Thanks to solutions based on artificial intelligence, one can protect from the trap of excessive debt | 2.81 | 3.00 | 2 | 1.030 | 1 | 5 |
| PU6. Solutions based on artificial intelligence allow to reduce the costs of using financial products and services | 3.41 | 4.00 | 4 | 1.016 | 1 | 5 |
| PU7. Solutions based on artificial intelligence can reduce the financial risk of consumers | 3.10 | 3.00 | 3 | 1.020 | 1 | 5 |
| PU8. AI-based solutions help to manage the saved money more effectively | 3.30 | 3.00 | 4 | 1.010 | 1 | 5 |
| PU9. AI-based solutions help protect against attempted financial fraud and money theft | 2.93 | 3.00 | 3 | 1.132 | 1 | 5 |
| PU10. AI-based solutions facilitate taxes settling | 3.44 | 4.00 | 4 | 1.052 | 1 | 5 |
| PU11. AI-based solutions help to reduce unnecessary expenses and save money | 3.26 | 3.00 | 4 | 0.983 | 1 | 5 |
| PU12. AI-based solutions allow to obtain quickly the financial data looked for | 3.67 | 4.00 | 4 | 1.090 | 1 | 5 |
| **Perceived ease of use** | | | | | | |
| PEOU1. Artificial intelligence provides consumers with a wide range of tools to make financial decisions | 3.46 | 4.00 | 4 | 0.997 | 1 | 5 |
| PEOU2. It's easy for consumers to choose the right AI-based tool to achieve their financial goals | 3.13 | 3.00 | 4 | 1.064 | 1 | 5 |
| PEOU3. It's easy for consumers to create questions/prompts for AI to get a satisfactory solution for their finance | 3.29 | 3.00 | 4 | 1.098 | 1 | 5 |
| PEOU4. It is easy for consumers to coordinate (combine) different IT tools so that AI executes different financial commands on their behalf | 3.15 | 3.00 | 3 | 1.095 | 1 | 5 |
| **Attitude towards AI** | | | | | | |
| ATT1. I am positively inclined about the use of artificial intelligence systems in the daily management of my own finances | 3.15 | 3.00 | 4 | 1.156 | 1 | 5 |
| ATT2. There are many beneficial applications of AI in financial management by consumers | 3.60 | 4.00 | 4 | 0.993 | 1 | 5 |
| ATT3. The prospect of implementing artificial intelligence in financial management by consumers is exciting | 3.21 | 3.00 | 4 | 1.068 | 1 | 5 |
| ATT4. I'm impressed by the possibilities that artificial intelligence offers to consumers for their financial management | 3.36 | 4.00 | 4 | 1.085 | 1 | 5 |
| ATT5. The use of artificial intelligence in consumers' financial management can have a positive impact on their financial well-being | 3.37 | 3.00 | 4 | 0.971 | 1 | 5 |
| ATT6. AI-based systems can help consumers feel happy | 3.00 | 3.00 | 3 | 1.095 | 1 | 5 |
| ATT7. AI-based systems can perform certain financial management (financial decision-making) tasks better than consumers | 3.29 | 3.00 | 4 | 1.076 | 1 | 5 |
| ATT8. A large part of society will benefit from the future thanks to their use of artificial intelligence in managing their own finances | 3.41 | 4.00 | 4 | 1.029 | 1 | 5 |
| ATT9. For routine financial transactions, I would rather use interaction with an AI-based system than with a human (advisor) | 2.78 | 3.00 | 2 | 1.264 | 1 | 5 |

*(Continued)*

**Table 2.** (Continued)

| Items | Mean | Median | Moderate | Std. | Min. | Max. |
|-------|------|--------|----------|------|------|------|
| ATT10. I believe that artificial intelligence in consumer financial management is safe when it is implemented responsibly | 3.36 | 3.00 | 4 | 1.112 | 1 | 5 |
| ATT11. I believe that allowing artificial intelligence by consumers to manage their own finances can be dangerous | 3.53 | 4.00 | 4 | 1.068 | 1 | 5 |
| ATT12. Consumers' use of artificial intelligence to manage their own finances causes them to be spied on | 3.31 | 3.00 | 3 | 1.065 | 1 | 5 |
| ATT13. I feel uncomfortable thinking about future use of artificial intelligence for financial management by consumers | 3.15 | 3.00 | 3 | 1.177 | 1 | 5 |
| ATT14. Artificial intelligence used by consumers to manage their own finance can take control of them | 3.13 | 3.00 | 4 | 1.180 | 1 | 5 |
| ATT15. I think that consumers' use of AI in managing their own finance leads them to many wrong financial decisions | 3.13 | 3.00 | 3 | 1.080 | 1 | 5 |
| ATT16. Consumers will increasingly suffer from the use of artificial intelligence in taking financial decisions | 2.99 | 3.00 | 3 | 1.073 | 1 | 5 |
| **Behavioral intention** | | | | | | |
| BI1. I'm going to start using artificial intelligence soon to manage my own finances (e.g., make financial decisions) | 2.61 | 2.00 | 2 | 1.129 | 1 | 5 |
| BI2. I will start using artificial intelligence in the indefinite future to manage my own finances (e.g., make financial decisions) | 2.82 | 3.00 | 2 | 1.098 | 1 | 5 |
| BI3. I will recommend others to use artificial intelligence to manage their finance (e.g., making financial decisions) | 2.55 | 2.00 | 2 | 1.073 | 1 | 5 |
| BI4. I intend to continue using artificial intelligence to manage my own finances (e.g., make financial decisions) | 2.67 | 3.00 | 2 | 1.112 | 1 | 5 |
| **Actual usage** | | | | | | |
| AU1. I use a solution based on artificial intelligence, which allows me to better manage my own budget (my own expenses, income) | 2.36 | 2.00 | 2 | 1.114 | 1 | 5 |
| AU2. I use an AI-based solution that helps me make trades in the stock market | 2.15 | 2.00 | 2 | 1.091 | 1 | 5 |
| AU3. I use a chat (virtual assistant) based on artificial intelligence to find answers to my questions in the field of finance | 3.12 | 3.00 | 4 | 1.336 | 1 | 5 |
| AU4. I use an AI-powered solution that helps prevent hackers from phishing my passwords and funds | 2.56 | 2.00 | 2 | 1.221 | 1 | 5 |

for the variables AU2 (2.15) and AU1 (2.36). This means that few people use AI-based tools as a support for financial decision-making. The median for most variables is 3. The variable PU1 has the lowest standard deviation (0.943), suggesting that responses to this variable were more closely aligned and the spread of responses was the smallest. In contrast, variables such as AU3 (1.336) have a higher standard deviation, indicating a greater diversity of respondents' opinions. All variables show the same range of responses, from 1 to 5, indicating that respondents used the entire available scale.

In addition to the individual variables' analysis, we prepared descriptive statistics for the groups identified according to the TAM model structure (Table 3) and to the scope of consumer finance management (Table 3). Such analysis allowed us to identify the highest-graded features of the use of AI in financial decision-making. The mean and median scores of responses presented in Table 1 indicate that the highest scores were achieved for the perceived usefulness of AI (3.52 and 3.33), followed by perceived ease of use (3.26 and 3.22) and attitudes toward the use of AI (3.24 and 3.25). The lowest gradings (2.55 and 2.46) were observed for actual use and behavioral intentions (2.66 and 2.64). The results may confirm that AI seems to be perceived as very useful in financial management, but not necessarily easily adopted and used in practice.

**Table 3. Descriptive statistics in TAM grading.**

| TAM scope grading | Mean | Median | Std. | Min | Max | Max-min |
|---|---|---|---|---|---|---|
| Perceived usefulness | 3.35 | 3.33 | 0.33 | 2.81 | 3.91 | 1.1 |
| Perceived ease of use | 3.26 | 3.22 | 0.15 | 3.13 | 3.46 | 0.33 |
| Attitudes toward use | 3.24 | 3.25 | 0.21 | 2.78 | 3.6 | 0.82 |
| Behavioral intention | 2.66 | 2.64 | 0.12 | 2.55 | 2.82 | 0.27 |
| Actual use | 2.55 | 2.46 | 0.42 | 2.15 | 3.12 | 0.97 |

Apart from the TAM, we regroup the data to calculate the descriptive statistics of AI importance in the specified financial use. According to the mean and median values, we could find that AI is particularly important in the aspect of financial literacy (3.63 and 3.74). AI was less graded in the scope of current financial management (3.10 and 3.21) and cyber and financial security (3.15 and 3.14). The least important, according to the respondents, was the implementation of AI for financial risk management (2.96 and 2.96).

## Results

Our investigation began by testing whether the construct measures meet the recommended guidelines for reliability and validity (Table 5). We tested the reliability of the variance using Cronbach's alpha and Composite Reliability (CR). Discriminant validity was used to determine if the construct was empirically distinct from other constructs in the structural model [72]. [73] propose the minimum acceptable value of Alpha and Composite reliability as 0.7. Validity assessment focuses on the convergent validity of each measure using the average variance extracted (AVE). The convergent validity of the construct is achieved when the average variance extracted is higher than 0.50. The results for reliability, validity, and factor loadings for the remaining items are included in Table 4. All Cronbach's alpha values, CR, and AVE are higher than recommended values. Although factor loadings below 0.7 are often considered insufficient and may be candidates for removal [72], we intentionally retained four items with loadings below this threshold. As highlighted by [72], such decisions should be approached with caution, considering their potential impact on other measures of construct validity and reliability. In our case, removing these items did not lead to any improvement in internal consistency, reliability, or convergent validity. Therefore, we opted to retain them in line with the recommended careful evaluation process. Additionally, [74] points out that factor loadings of 0.5 or 0.6 can still be acceptable, particularly when other indicators in the construct provide a comparative basis. Beyond statistical justification, these items – PU7, PU8, PU12, and ATT7 – were retained due to their theoretical importance and unique conceptual contribution. They capture key facets of how consumers perceive AI-based tools: their potential to reduce financial risk, improve savings management, enable fast access to financial data, and outperform human judgment in specific financial tasks. These dimensions are central to understanding the broader constructs of perceived usefulness and attitudes toward AI in finance. Excluding them would have weakened the content validity of the measurement model and narrowed the conceptual scope of the study. The last column of Table 5 presents variance inflation factors (VIF). It describes the degree of collinearity between the loadings. The smallest possible value

**Table 4. Descriptive statistics in AI-based tools designation.**

| Finance scope grading | Mean | Median | Std. | Min | Max | Max-min |
|---|---|---|---|---|---|---|
| Cyber and financial security | 3.15 | 3.14 | 0.28 | 2.56 | 3.53 | 0.97 |
| Financial literacy | 3.63 | 3.74 | 0.35 | 3.12 | 3.91 | 0.79 |
| Current finance management | 3.10 | 3.21 | 0.36 | 2.36 | 3.60 | 1.24 |
| Investments and savings | 3.01 | 3.25 | 0.50 | 2.15 | 3.37 | 1.22 |
| Financial risk management | 2.96 | 2.96 | 0.21 | 2.81 | 3.10 | 0.29 |

**Table 5. Items loadings, reliability, and validity.**

| Variables | Abbreviations | λ | VIF |
|---|---|---|---|
| Perceives usefulness of AI in finance management (PU) Cronbach's alpha = 0.842; CR = 0.884; AVE = 0.515 | PU1 | 0.729 | 1.708 |
| | PU2 | 0.798 | 2.270 |
| | PU3 | 0.752 | 1.982 |
| | PU6 | 0.721 | 1.685 |
| | PU7 | 0.685 | 1.537 |
| | PU8 | 0.645 | 1.437 |
| | PU12 | 0.680 | 1.521 |
| Perceived ease of use AI in finance management (PEOU) Cronbach's alpha = 0.806; CR = 0.828; AVE = 0.718 | PEOU2 | 0.875 | 1.712 |
| | PEOU3 | 0.830 | 1.725 |
| | PEOU4 | 0.837 | 1.810 |
| Attitude towards AI in finance management (ATAI) *Cronbach's alpha = 0.891; CR = 0.894; AVE = 0.570* | ATT1 | 0.756 | 1.911 |
| | ATT2 | 0.800 | 2.271 |
| | ATT3 | 0.703 | 1.786 |
| | ATT4 | 0.786 | 2.126 |
| | ATT5 | 0.803 | 2.210 |
| | ATT7 | 0.687 | 1.763 |
| | ATT8 | 0.764 | 2.061 |
| | ATT10 | 0.731 | 1.787 |
| Behavioral intention related to using AI in finance management (BI) Cronbach's alpha = 0.849; CR = 0.851; AVE = 0.768 | BI1 | 0.897 | 2.410 |
| | BI2 | 0.898 | 2.462 |
| | BI3 | 0.834 | 1.729 |
| Actual usage of AI tools in finance management Cronbach's alpha = 0.768; CR = 0.789; AVE = 0.593 | AU1 | 0.853 | 1.915 |
| | AU2 | 0.832 | 2.020 |
| | AU3 | 0.652 | 1.245 |
| | AU4 | 0.727 | 1.510 |

for the VIF is 1, which indicates the complete absence of collinearity. Typically, in practice, there is a small amount of collinearity among predictors. Collinearity issues are usually uncritical if VIF takes values from 3 to 5 [72]. In our analysis, the VIF values didn`t exceed the value 3. It means that the collinearity is not an issue in our case.

Apart from the above-presented evaluation, we additionally tested discriminant validity using cross-factor loading and the Heterotrait-Monotrait Method (HTMT). Table 6 shows the cross-factor loadings for all the items. Our analysis shows that the individual loadings (observed variables) are correctly assigned to the latent variables. The loadings are specific to the single constructs they comprise and do not influence more than one construct.

Table 7 shows discriminant validity using the Heterotrait-Monotrait Method (HTMT) developed by [75]. The HTMT is defined as the mean value of the item relations across constructs relative to the (geometric) mean of the average relations for the items measuring the same construct [76]. In our analyses, all pairs of latent variables fall within the stricter approach – the HTMT value does not exceed 0.85 [75].

To estimate the model fit measurement, we used the Standardized Root Mean Square Residual (SRMR). The SRMR is the difference between the observed correlation and the model-implied correlation matrix. Thus, it allows assessing the average magnitude of the discrepancies between observed and expected correlations as an absolute measure of (model) fit criterion. The value for the estimated model is 0.077, which falls within the acceptable range below 0.08, indicating that the model closely fits the empirical data [77]. A good model fit indicates a close approximation between the model and the sample. The structural model in partial least squares regression can be evaluated using an adjusted coefficient of

**Table 6. Discriminant validity – cross factors loading.**

| | PU | PEOU | ATT | BI | AU |
|---|---|---|---|---|---|
| PU1 | **0.729** | 0.349 | 0.509 | 0.381 | 0.226 |
| PU2 | **0.798** | 0.392 | 0.506 | 0.321 | 0.120 |
| PU3 | **0.752** | 0.340 | 0.447 | 0.258 | 0.094 |
| PU6 | **0.721** | 0.311 | 0.451 | 0.215 | 0.071 |
| PU7 | **0.685** | 0.332 | 0.421 | 0.325 | 0.205 |
| PU8 | **0.645** | 0.308 | 0.433 | 0.321 | 0.149 |
| PU12 | **0.680** | 0.401 | 0.456 | 0.273 | 0.086 |
| PEOU2 | 0.495 | **0.875** | 0.432 | 0.350 | 0.262 |
| PEOU3 | 0.377 | **0.830** | 0.340 | 0.224 | 0.193 |
| PEOU4 | 0.342 | **0.837** | 0.351 | 0.264 | 0.289 |
| ATT1 | 0.507 | 0.318 | **0.756** | 0.544 | 0.355 |
| ATT2 | 0.574 | 0.336 | **0.800** | 0.357 | 0.288 |
| ATT3 | 0.341 | 0.296 | **0.703** | 0.445 | 0.343 |
| ATT4 | 0.462 | 0.399 | **0.786** | 0.405 | 0.349 |
| ATT5 | 0.540 | 0.297 | **0.803** | 0.461 | 0.337 |
| ATT7 | 0.482 | 0.320 | **0.687** | 0.368 | 0.287 |
| ATT8 | 0.512 | 0.405 | **0.764** | 0.328 | 0.281 |
| ATT10 | 0.449 | 0.329 | **0.731** | 0.420 | 0.307 |
| BI1 | 0.362 | 0.311 | 0.486 | **0.897** | 0.560 |
| BI2 | 0.420 | 0.241 | 0.513 | **0.898** | 0.451 |
| BI3 | 0.323 | 0.336 | 0.454 | **0.834** | 0.512 |
| AU1 | 0.104 | 0.203 | 0.321 | 0.528 | **0.853** |
| AU2 | 0.085 | 0.251 | 0.251 | 0.467 | **0.832** |
| AU3 | 0.322 | 0.265 | 0.473 | 0.406 | **0.652** |
| AU4 | 0.103 | 0.186 | 0.269 | 0.361 | **0.727** |

**Table 7. Discriminant validity using the Heterotrait-Monotrait Method (HTMT).**

| | ATT | AU | BI | PEOU | PU |
|---|---|---|---|---|---|
| ATT | | | | | |
| AU | 0.518 | | | | |
| BI | 0.635 | 0.711 | | | |
| PEOU | 0.521 | 0.374 | 0.400 | | |
| PU | 0.738 | 0.247 | 0.494 | 0.577 | |

determination $R^2$. It measures the variance, which is explained in each of the endogenous constructs and is, therefore, a measure of the model's explanatory power. The $R^2$ ranges from 0 to 1, with higher values indicating a greater explanatory power [76]. In our model, $R^2$ for the perceived usefulness of AI was 0.235, for the behavioral intention was 0.309, for the actual usage of AI tools 0.335, and $R^2$ for attitude toward AI in consumer finance management reached 0.436.

Table 8 shows the verified hypotheses on the direct relationships between the analyzed variables. The effects of perceived ease of use of AI-based tools for financial decision-making on the perceived usefulness and on the attitude toward AI are statistically significant: PEOU→PU ($\beta = 0.487$, t = 10.329, $p < 0.001$), PEOU→ATT ($\beta = 0.173$, t = 3.265, $p = 0.001$). Thus, our hypotheses $H_1$, $H_2$ were supported. Likewise, the effects of perceived usefulness on attitude toward

**Table 8. The list of verified hypotheses.**

|  |  | β | t | p |
|---|---|---|---|---|
| H1: | PEOU→PU | 0.487 | 10.329 | 0.000 |
| H2: | PEOU→ATT | 0.173 | 3.265 | 0.001 |
| H3: | PU→ATT | 0.560 | 10.055 | 0.000 |
| H4: | PU→BI | 0.108 | 1.518 | 0.129 |
| H5: | ATT→BI | 0.485 | 7.240 | 0.000 |
| H6: | BI→AU | 0.581 | 14.253 | 0.000 |
| H7: | PEOU→PU→ATT | 0.273 | 6.694 | 0.000 |
| H8: | PU→ATT→BI | 0.272 | 5.201 | 0.000 |

AI are statistically significant: PU→ATT ($\beta=0.560$, t=10.055, $p<0.001$). The hypothesis $H_3$ was supported. In contrast, the hypothesis $H_4$ was rejected. Behavioral intention of the use of AI tools supporting financial decisions is not affected by the perceived usefulness: PU→BI ($\beta=0.108$, t=1.518, $p=0.129$). According to our modeling, attitudes toward AI positively impact the behavioral intentions of use: ATT→BI ($\beta=0.485$, t=7.240, $p =<0.001$). Thus, hypothesis $H_5$ was supported. Similarly, behavioral intentions clearly support actual use of AI: BI→AU ($\beta=0.581$, t=14.253, $p<0.001$), which positively verifies hypothesis $H_6$. The last two hypotheses concerned the mediating role of perceived usefulness between perceived ease of use and attitude toward AI tools, as well as the attitude toward AI tools mediating perceived usefulness and behavioral intention. Perceived ease of use and attitudes regarding the use AI tools were positive and statistically significant: PEOU→PU→ATT ($\beta=0.273$, t=6.694, $p<0.001$) as the perceived usefulness and behavioral intention to use AI: PU→ATT→BI ($\beta=0.272$, t=5.201, $p<0.001$). Hypotheses $H_6$ and $H_7$ were then supported. In the case of first mediation, PEOU→PU→ATT, the indirect effect and the direct effect are significant. Such a case, according to [78], represents complementary mediation (partial mediation). In case of second mediation PU→ATT→BI, the indirect effect is statistically significant, but the direct effect is not. Such a case represents only indirect mediation (full mediation).

Apart from the significant relationships, we also examined the effect size of the paths ($f^2$) [59]. The effect size is a measure used to assess the relevant impact of a predictor construct on an endogenous construct. According to [79], the relationships PU→ATT ($f^2=0.426$), PEOU→PU ($f^2=0.310$), BI→AU ($f^2=0.507$), ATT→BI ($f^2=0.197$) represent a large effect size. Meanwhile, the relationship PEOU→ATT ($f^2=0.041$) had a small effect size.

To deepen our investigation of the relations between the constructs, i.e., latent variables from PLS-SEM modelling, we conducted Importance-Performance Map Analysis (IPMA). The IPMA identifies the priority of individual factors (variables) determining the specified construct. The importance-performance map analysis cross-examines the performance and importance of the item in its impact on the specified construct. The objective is to identify the unstandardized total effect of the individual variables of predeceasing constructs (e.g., PU, PEU, ATT, BI) in anticipating a specific target endogenous construct (e.g., actual usage). The total effect demonstrates the importance and performance of apparent variables, where the performance is set as the mean value of their scores (ranging from 0 – the lowest to 100 – the highest). The interpretation of IPM analysis is that a 1-unit increase in the predecessor's performance increases the performance of the target construct by the size of the predecessor's unstandardized total effect. The analysis determines four structures, i.e., a) factors of high importance and performance – they are critical for the success of the impact and are performing well. These are the strengths to be maintained; b) factors of low importance and high performance – they are less critical in the impact analysis and to develop an investigated construct, it might be possible to allocate fewer resources; c) factors of low importance and low performance – they do not bring a significant impact in the construct; d) factors of high importance and low performance – they constitute the critical improvement area; these factors are important but underperforming, thus their improvement will influence the endogenous, specific construct.

The fundamental aspect of the IPMA is to identify predecessors that have relatively high importance for the target (endogenous) construct (i.e., those that have a strong total effect) but also have relatively low performance (i.e., low average latent variable scores) [77]. Their improvement is then critical to improve the target construct results. Such an idea is particularly useful for managers or other decision-makers who tend to achieve specified goals (e.g., the number of AI products used).

Fig 2 presents the Importance-Performance Map, which contains variables influencing the actual use of AI tools in personal finance. In the graph, all the variables BI are of high importance but low performance. They are also clearly distant from other variables representing other constructs (PU, PEU, ATT). To increase the level of use of AI solutions, the key stakeholders should first enhance the performance of the behavioral construct, i.e., take actions to increase behavioral intentions among potential users.

As the actual use of AI-based solutions turned out to be determined strictly by the BI construct and all three BI variables, the next step in our research was to identify the factors that support the behavioral intention to use AI. For this purpose, we performed a second IPMA related to behavioral intention. Fig 3 presents individual variables influencing the behavioral intention to use AI solutions. The map indicates that perceived ease of use loadings PEOU2, PEOU3, and PEOU4 have the strongest impact on behavioral intentions to use AI. All of them are also of low performance. Supporting potential users in understanding the ease of use of AI solutions, particularly in constructing prompts and coordinating different ICT tools for AI use, should be considered crucial in achieving a more substantial impact on their behavioral intentions and actual use of AI. The interesting result is that individual variables related to the perceived usefulness of AI tools in supporting financial decision-making showed high performance but little impact on the behavioral intention and actual use of the AI, as visible in Figs 2 and 3.

## Discussion

The results of our investigation support several traditional TAM assumptions, but simultaneously they reveal notable divergences from the adoption patterns reported in previous studies on digital financial tools and AI-based solutions, including robo-advisory services. The results outline a coherent structural framework in which both direct and mediated relationships within the proposed TAM jointly shape consumers' acceptance and use of AI-based tools in making decisions in the scope of personal finance.

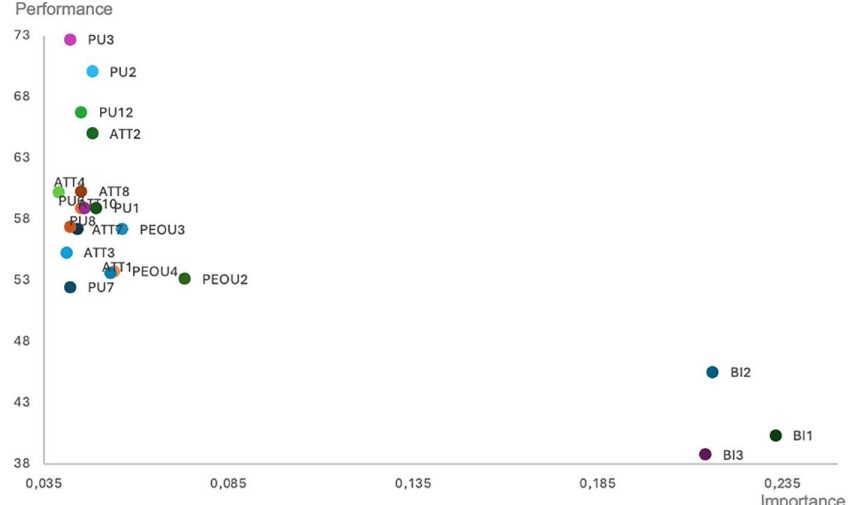

**Fig 2. IPMA for predecessor variables determining the actual use of AI.**

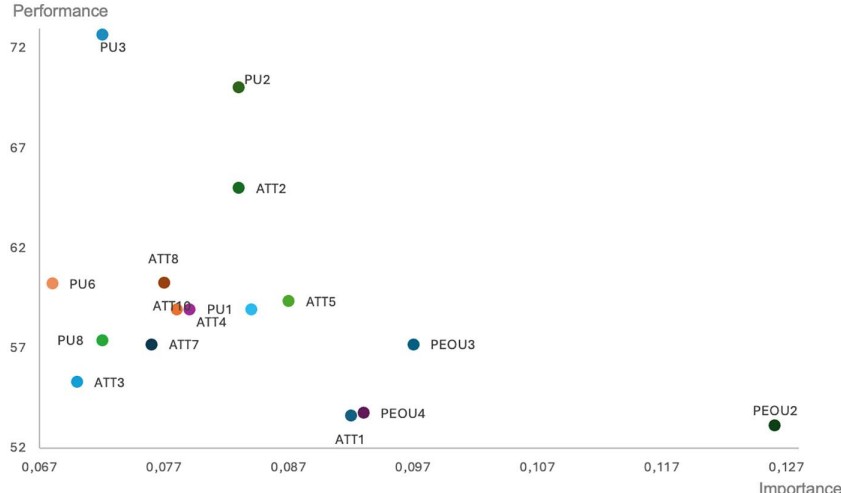

**Fig 3. IPMA for predecessor variables determining the behavioral intentions to use AI.**

The analysis of descriptive statistics in TAM grading, as well as the designation of AI-based tools, demonstrates the perception of AI primarily through its usefulness and its use for increasing financial literacy. Such responses may suggest that AI, as an emerging, innovative, and game-changing solution in many sectors, is also viewed in a similar way in finance. The financial literacy priority may suggest a willingness to utilize it as a professional advisor and source of adequate and timely financial information, which can be considered a soft impact of AI on personal financial decisions.

Our first hypothesis, assuming that perceived ease of use positively influences the perceived usefulness of AI-based tools in personal financial decision-making, was confirmed. This result demonstrates that when users find AI solutions intuitive and straightforward, they are more likely to perceive them as genuinely useful for supporting financial tasks. The relationship between perceived ease of use and usefulness emerged as an important component of our model, consistent with prior research on digital and AI-based financial technologies [21,53,55,57,58]. In the context of AI-based financial tools, this mechanism is particularly distinctive because the effective use of AI solutions requires not only basic interaction skills but also the ability to formulate effective prompts, select appropriate applications or AI agents, and coordinate multiple digital resources. Consequently, a higher sense of simplicity makes users more inclined to perceive AI-based tools as practical and relevant in managing their finances.

Regarding the second hypothesis, which assumed a positive relationship between perceived ease of use and attitudes toward AI-based tools used in personal financial decision-making, our results revealed a positive but weak association. Similar modest associations have been reported in studies on robo-advisory services [20] and on bio-QR payment technologies [56], indicating that ease of use plays a limited role in shaping attitudes toward various emerging financial technologies. We can then conclude that, even if AI-based solutions are found as simple and straightforward, they do not significantly shape a positive attitude toward AI.

The third hypothesis, assuming a positive relationship between perceived usefulness and attitudes toward AI-based tools used in personal financial decision-making, was also confirmed by our results. This relationship was one of the strongest in the model, indicating that users' attitudes are motivated primarily by how useful they perceive such tools. Strong associations between usefulness and attitudes have also been reported in studies on robo-advisory services [20] and in research on bio-QR payment technologies [56], confirming the central role of perceived usefulness in shaping the positive attitudes across different emerging financial technologies.

One of the most significant findings of our study is undoubtedly the rejection of the fourth hypothesis, which assumed a direct positive relationship between perceived usefulness and behavioral intention to use AI-based tools for personal financial decision-making. This result contrasts with evidence from several established financial technologies, such as mobile banking, QR-code payments, and electronic wallets, where perceived usefulness has consistently been shown to directly predict behavioral intention [54,58,59]. On the other hand, the findings regarding AI-related financial tools are mixed. For example, some studies on robo-advisory services reported a significant PU→BI effect [21], whereas others found no such relationship [20]. Our results follow the latter pattern and are fully aligned with the mediation analysis, which revealed a full mediation effect PU→ATT→BI, thereby confirming hypothesis H8. Perceived usefulness affects the behavioral intention exclusively through attitudes. This pattern is consistent with the observations of Kim et al. [80], who showed that in certain TAM-based configurations, attitudes can serve as the primary transmission channel through which usefulness affects intention. The convergence of findings across two AI-based financial domains – our study on AI usage in personal financial decisions and previous research on robo-advisory services [20,21] – suggests that attitudes may serve as a central decision-making mechanism, irrespective of whether an AI system simply supports users' decisions or operates with greater autonomy. This mechanism can be further understood in light of prior work, which shows that the use of AI in financial decision-making processes may entail higher consequential risk, as algorithmic outputs can have direct implications for users' financial outcomes [20]. As shown in broader empirical research, in contexts characterized by perceived risk, affective evaluations tend to mediate the relationship between perceived benefits and behavioral intention [81]. Consequently, perceived usefulness may not be sufficient to generate an intention to use unless it first contributes to the formation of a favorable overall attitude toward the technology. Such a situation may arise particularly when a technology is not yet well established in practice, as is currently the case with AI, and when prejudices, misconceptions, and myths lead to binary and vague perceptions of its risks, dangers, and potential benefits [82]. Taken together, these findings outline a consistent pattern in which attitudes serve as a pivotal mechanism linking perceptions of usefulness to downstream behavioral outcomes. Concerning the fifth hypothesis, our results confirmed a strong positive relationship between attitudes toward AI-based tools and behavioral intention to use them. This finding is fully consistent with the core assumptions of TAM, in which attitudes function as a proximal determinant of behavioral intention, and aligns with previous research on digital and AI-enabled financial technologies showing that favorable attitudes are a key driver of technology adoption [20,21,53,58]. The strength of this association underscores the pivotal role of attitudinal evaluations in influencing consumers' willingness to use AI when making personal financial decisions.

The sixth hypothesis, predicting a positive relationship between behavioral intention and actual use, was also supported. This result confirms one of the most established findings in technology acceptance research, namely that intention serves as a reliable predictor of subsequent usage behavior. Evidence from prior work on digital financial technologies confirms this linkage, showing that behavioral intention constitutes a key antecedent of actual adoption [65].

Our results also confirm the seventh hypothesis, that the impact of ease of use on attitudes is driven primarily by perceived usefulness. In our model, the indirect pathway through perceived usefulness accounted for a considerably larger share of the overall effect than the direct link. A very similar pattern was reported in the context of robo-advisory services [20], where perceived usefulness acted as the dominant mechanism connecting ease of use with attitudinal evaluations. Taken together, these findings suggest that in AI-based financial technologies, perceived usefulness consistently serves as the primary channel through which ease of use influences users' attitudes.

By supplementing the PLS-SEM modelling with an in-depth Importance–Performance Map Analysis, we were able to identify the elements with the greatest practical impact on the actual use of AI-based tools. The IPMA results showed that loadings of behavioral intention had the highest importance for explaining actual use, yet their performance remained relatively low. This suggests that, although users recognize the value and potential of AI-based tools in supporting financial decisions, their willingness to consistently and meaningfully engage with these tools is limited. Such hesitation may stem

 

from a lack of knowledge about effective AI use, concerns about reliability, or a shortage of educational resources that would facilitate the integration of AI into personal finance practices.

A similar pattern was observed for behavioral intention itself. The loadings of perceived ease of use included in the model demonstrated high importance for shaping intention, yet their performance was also low. This suggests that respondents perceive ease of use as a crucial prerequisite for forming the intention to use AI-based tools; however, in practice, they often encounter difficulties when interacting with such technologies. These challenges likely arise from the need to develop new competencies — such as formulating prompts, selecting appropriate functions, and coordinating multiple AI-based applications — within a rapidly evolving technological environment.

Taken together, the high importance but low performance of both behavioral intention and perceived ease-of-use loadings indicate notable barriers that must be addressed to enhance the adoption of AI-based tools for decision-making in personal finance management. Supportive measures such as user education, guided onboarding, interactive tutorials, and low-risk experimentation training may help reduce these barriers and improve overall performance, ultimately enabling more effective and confident use of AI in financial decision-making. These dynamics are further reflected in how consumers respond to currently available AI-based financial tools.

Although our study did not analyze specific AI applications, several consumer-facing tools provide a relevant backdrop for interpreting these results. For example, Finchat supports retail investors with algorithm-based investment guidance; TurboTax Assistant offers AI-driven assistance during tax filing; tools such as Cleo and Emma help users manage expenses and monitor recurring payments. Applications like Acorns, Qapital, and Chip automate saving and entry-level investing. The presence of such tools highlights the increasing role of AI in personal finance management, yet their varied adoption levels suggest that the perceived functional value does not automatically translate into actual use. The full mediation observed between perceived usefulness and behavioral intention in our model helps explain this discrepancy: a positive attitude toward AI-based financial tools appears to be a necessary condition for converting perceived utility into real engagement. Taken together, these examples illustrate how the psychological mechanisms identified in our model translate into observable patterns of consumer engagement in AI-based financial solutions.

## Conclusions

This study offers empirical insights into AI-based tools that aid consumers in making financial decisions, an area that remains underexplored in the technology acceptance literature. It contributes to ongoing discussions on how consumers evaluate and accept AI systems that support, rather than automate, financial decisions, offering implications for theory, practice, and policy. Moreover, our study verified the validity, internal consistency, and logical coherence of the questionnaire we developed, confirming that the items were clearly understood and appropriately interpreted by respondents. Our findings demonstrate that perceived usefulness influences behavioral intention to use AI-based solutions through users' overall attitudes, underscoring the central role of attitudinal evaluation when individuals consider adopting AI-based decision-support tools. Understanding the acceptance of AI systems is becoming increasingly important, as consumers face growing exposure to misleading financial information, online scams, and phishing attempts. This makes reliable decision-support tools potentially valuable in everyday financial decision-making. The limited knowledge of finance and financial investment among society also provides justification for AI-based solutions to support more efficient long-term strategic financing and investing decisions of consumers.

### Theoretical implications

The findings indicate that attitude is the main channel through which perceived usefulness influences behavioral intention. Rather than exerting a direct effect on intention, perceived usefulness operates primarily through users' overall evaluations of the tool in this decision-support context. It suggests that the acceptance of AI-based tools depends more strongly on users' overall evaluation when the technology supports consumers in their financial decision-making rather

than making decisions autonomously. The study, therefore, highlights that partially autonomous AI-driven tools form a distinct category within the realm of financial technologies. Unlike robo-advisory systems, which replace parts of the decision-making process, the AI solutions examined by us serve as a decision-support mechanism, allowing the user to retain full control. This difference in technological role appears to alter the adoption pathway and may require refinement of the existing acceptance models. By examining this intermediate form of AI-driven financial technology, the study contributes to the growing body of research on how consumers evaluate AI systems that augment rather than replace human decision-making.

## Managerial implications

Our findings indicate that improving functional performance alone is insufficient to increase adoption of AI-based financial tools. Because attitudes constitute the primary pathway to behavioral intention, practical efforts should prioritize strengthening users' clarity and perceived control when interacting with such systems. Ease-of-use–related activities, such as formulating prompts, selecting system functions, and coordinating multiple features, represent key friction points. Developers should therefore focus on usability-oriented solutions, including simplified interfaces, guided workflows, and example prompts, supported by structured onboarding that demonstrates how system features work together. Providing concise, task-specific instructions for operating core functions can further support efficient use and strengthen perceived ease of use. From a policy perspective, the findings underscore the importance of integrating AI literacy into consumer financial-education initiatives. Here, the emphasis should not be on operating the AI tools themselves but on helping individuals understand the appropriate role of AI in financial decision-making, including its limitations, typical use cases, and the distinction between supportive and autonomous functions. Policy measures that promote clear system presentation and responsible deployment standards can further ensure informed and effective use of AI-based financial tools.

## Limitations

This study has several limitations. The sample consists primarily of young adults with relatively high digital competence, which may limit the generalizability of the findings to older or less technologically experienced populations. Additionally, the cross-sectional design captures perceptions at a single point in time and does not account for potential changes as AI systems evolve or as users gain experience. Finally, reliance on self-reported data may not fully reflect actual behavior in real financial settings.

## Ideas for future research

Future research may address the limitations and broaden the understanding of the acceptance of AI-based tools in personal finance management. First, studies involving more diverse and representative populations could clarify whether the psychological mechanisms identified in this research apply across age groups, socioeconomic segments, and varying levels of financial and technological proficiency. Second, extending the model with additional external variables – such as financial literacy, ICT competence, effort expectancy, performance expectancy, and self-development inclination – may improve understanding of individual differences in perceptions and attitudes toward AI-based financial tools. Third, longitudinal or time-sensitive research designs would help determine how acceptance and attitudes evolve over time as users gain experience with AI systems. Such designs could reveal whether the attitudinal pathway identified here remains stable or changes as users become more familiar with AI-based decision-support tools. Fourth, future research could examine the role of AI-based decision-support tools in helping consumers navigate the growing prevalence of online financial fraud, scams, and phishing attempts. As individuals are increasingly exposed to misleading or deceptive information, understanding whether and how AI assistance can support verification or reduce errors represents an important direction for further investigation.

## Author contributions

**Conceptualization:** Tomasz Szopiński, Michal Buszko.

**Data curation:** Tomasz Szopiński.

**Funding acquisition:** Michal Buszko.

**Investigation:** Michal Buszko.

**Methodology:** Tomasz Szopiński.

**Resources:** Michal Buszko, Małgorzata Porada-Rochoń.

**Software:** Tomasz Szopiński.

**Supervision:** Michal Buszko, Małgorzata Porada-Rochoń.

**Visualization:** Tomasz Szopiński.

**Writing – original draft:** Tomasz Szopiński, Michal Buszko, Małgorzata Porada-Rochoń.

**Writing – review & editing:** Michal Buszko, Małgorzata Porada-Rochoń.

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
