## [Decision Letter · Decision Letter 0]

21 Apr 2025

Dear Dr. Buszko,

Thank you for submitting your manuscript to PLOS ONE. After careful consideration, we feel that it has merit but does not fully meet PLOS ONE’s publication criteria as it currently stands. Therefore, we invite you to submit a revised version of the manuscript that addresses the points raised during the review process.

**ACADEMIC EDITOR:**

Please address all the comments and suggestions given by the reviewers.

We look forward to receiving your revised manuscript.

Kind regards,

Kashif Ali, PH.D

Academic Editor

PLOS ONE

Journal Requirements:

Reviewers' comments:

Reviewer's Responses to Questions

**Comments to the Author**

1. Is the manuscript technically sound, and do the data support the conclusions?

Reviewer #1: Yes

Reviewer #2: Yes

2. Has the statistical analysis been performed appropriately and rigorously?

Reviewer #1: Yes

Reviewer #2: Yes

3. Have the authors made all data underlying the findings in their manuscript fully available?

Reviewer #1: Yes

Reviewer #2: Yes

4. Is the manuscript presented in an intelligible fashion and written in standard English?

Reviewer #1: Yes

Reviewer #2: Yes

Reviewer #1: Thank you for giving me the opportunity to review this paper.

Novelty and originality of the research must be added to the abstract.

A good informative abstract acts as a surrogate for the work itself. The researcher presents and explains the paper’s main arguments, significant results, and evidence. An informative abstract includes the information found in a descriptive abstract [purpose, methods, scope]. However, it also consists of a judgment or comment about the study’s validity, reliability, or completeness, the results and conclusions of the research, and the author’s recommendations.

The introduction is long.The introduction should clearly illustrate (1) what we know (the key theoretical perspectives and empirical findings) and what do we not know (major, unaddressed puzzle, controversy, or paradox does the study addresses, or why it needs to be addressed and why this matters). And, (2) what will we learn from the study and how does the study fundamentally change, challenge, or advance scholars’ understanding. Much sharper problematization is required so that the introduction draws the reader into the paper. The introduction therefore needs to do a better job in setting the stage for the articulation of the theoretical contributions of the study. At the end of the introduction, we should have a clear idea of what the paper is about (i.e. its motivation, the gap in understanding that the paper is trying to address and summary of theoretical contributions).

The literature review must engage in the constructs of your analytical framing in a meaningful way. The literature review section could be improved by being more analytical. In other words, building on the existing literature to highlight what is missing and what is yet to be done and in so doing outline the theoretical puzzles or debates to which this work contributes. I have concerns related to theoretical development, and note the need for a more rigorous critique of the literature to help deepen the theoretical underpinnings of the study.

The Discussion lacks a critical synthesis and comparison. The Discussion should include a critical synthesis and comparison of the data with the literature. The discussion clearly explains how your study advanced the reader’s understanding of the research problem from where you left them at the end of your review of prior research.

The Conclusion does not adequately discuss the theoretical and managerial implications of the study. Summarize your thoughts and convey the larger significance of your research. Identify and discuss how a gap in the literature has been addressed and demonstrate the importance of your ideas. Introduce possible new or expanded ways of thinking about the research problem.

Also, state the ideas for future research in the conclusion. Make sure you create 3 subsections in the Conclusion: 1) Theoretical Implications, 2) Managerial or Policy Implications, and 3) Ideas for Future Research.

Thank you

Reviewer #2: ### Peer Review of Manuscript PONE-D-25-08396: "Acceptance of AI-based tools in consumer financial decision-making: An application of the extended Technology Acceptance Model"

#### Overview

This manuscript explores the acceptance of AI-based tools in consumer financial decision-making, using an extended version of the Technology Acceptance Model (TAM). The study, conducted on 371 respondents from three Polish universities, employs Partial Least Squares Structural Equation Modeling (PLS-SEM) to examine factors influencing the adoption of AI tools, with a focus on perceived usefulness, ease of use, attitudes, behavioral intentions, and actual use. The research addresses a gap in the literature by focusing on AI tools that support decision-making rather than automated systems like robo-advisors. While the study is well-structured and tackles a relevant topic, it has several weaknesses that need addressing to enhance its academic rigor and impact.

#### Weaknesses

The manuscript has several areas that require improvement to meet the standards of a high-quality publication:

1. **Limited Sample Representativeness**:

- The sample is heavily skewed toward young respondents (89.2% aged 18–24, primarily students) from three Polish universities, which limits the generalizability of findings. The manuscript acknowledges this as a pilot study but does not sufficiently discuss how this demographic bias impacts the results.

- The use of the CAWI (Computer-Assisted Web Interview) technique excludes individuals who are not regular users of electronic devices, further narrowing the sample’s representativeness.

- **Suggestion**: Expand the sample in future studies to include diverse age groups, educational backgrounds, and technological proficiency levels. Discuss how the young, tech-savvy sample may inflate positive perceptions of AI tools compared to the general population.

2. **Superficial Literature Review**:

- The literature review provides a broad overview of AI in finance but lacks depth in discussing prior studies on consumer acceptance of AI tools for decision-making. For example, it mentions robo-advisory studies but does not critically engage with their findings to highlight differences with the current study’s focus.

- Key concepts, such as AI literacy or trust in AI, are introduced briefly but not explored in depth, despite their relevance to consumer adoption.

- **Suggestion**: Strengthen the literature review by including more studies on AI in personal finance, particularly those addressing decision-making support. Discuss how trust, privacy concerns, or cultural factors influence AI adoption, and position the study more clearly within this context.

3. **Measurement Instrument Concerns**:

- The manuscript uses an adapted version of the General Attitudes Towards Artificial Intelligence Scale (GAAIS) and a self-developed questionnaire. However, the process of adapting and validating these instruments is not detailed, raising concerns about their suitability for the financial decision-making context.

- Several items have factor loadings below 0.7 (e.g., PU8, PU7, ATT7, AU3), which the authors retain without strong justification. This weakens the construct validity.

- The large number of items (e.g., 12 for perceived usefulness, 16 for attitudes) may lead to respondent fatigue, potentially affecting data quality.

- **Suggestion**: Provide a detailed explanation of the questionnaire development and validation process. Justify the retention of low-loading items with stronger theoretical or empirical reasoning. Consider reducing the number of items to improve survey efficiency and respondent engagement.

4. **Methodological Gaps**:

- The data collection period (February 8, 2024, to October 29, 2024) spans nine months, but the manuscript does not discuss whether external factors (e.g., changes in AI technology or economic conditions) during this period could have influenced responses.

- The study does not report response rates or discuss potential non-response bias, which is critical for survey-based research.

- The PLS-SEM model fit is assessed using SRMR and R², but other fit indices (e.g., NFI, GFI) are not reported, limiting the evaluation of model quality.

- **Suggestion**: Discuss the implications of the long data collection period and any external factors that may have affected responses. Report response rates and address non-response bias. Include additional fit indices to provide a comprehensive assessment of model fit.

5. **Underdeveloped Discussion**:

- The discussion section reiterates the results without sufficiently engaging with their theoretical or practical implications. For example, the finding that perceived usefulness does not directly influence behavioral intention (H4 rejected) is surprising but not explored in depth.

- The manuscript does not compare its findings with prior TAM studies in finance (e.g., mobile banking, robo-advisors) to highlight similarities or differences.

- The implications for financial literacy, a key finding, are mentioned briefly but not connected to broader educational or policy strategies.

- **Suggestion**: Expand the discussion to interpret unexpected findings (e.g., H4 rejection) and compare results with prior TAM studies. Discuss how the emphasis on financial literacy can inform AI tool design or educational initiatives.

6. **Overgeneralization of Findings**:

- The manuscript occasionally overstates the implications of its findings, given the pilot nature of the study and the non-representative sample. For example, claims about policymakers’ actions (e.g., improving ease of use) are broad and not sufficiently grounded in the study’s context.

- The cultural context of Poland is not discussed, despite potential differences in AI adoption compared to other countries.

- **Suggestion**: Qualify the findings by emphasizing the pilot nature of the study and the specific context (Poland, young respondents). Discuss how cultural or economic factors in Poland may influence results and suggest cross-cultural studies for future research.

7. **Clarity and Writing Issues**:

- The manuscript contains minor language errors and inconsistencies, such as the use of “PFD” (presumably personal financial decision-making) without definition, and the Polish phrase “Rozwiązania bazujące na sztucznej inteligencji” (PU10) in an otherwise English table.

- Some sections, particularly the introduction and discussion, are verbose and could be more concise to improve readability.

- The figures (e.g., IPM maps) are referenced but not fully described in the text, making it difficult to interpret them without accessing the supporting files.

- **Suggestion**: Proofread the manuscript to correct language errors and ensure consistency (e.g., define acronyms, remove non-English text). Streamline verbose sections and provide detailed descriptions of figures in the main text.

8. **Limited Exploration of Socio-Demographic Factors**:

- The manuscript provides descriptive statistics for age, gender, and education but does not analyze how these factors influence AI adoption. Given the skewed sample, this is a missed opportunity to explore heterogeneity in responses.

- **Suggestion**: Conduct subgroup analyses (e.g., by gender or education level) to identify differences in AI acceptance. Discuss how socio-demographic factors may shape perceptions and intentions.

#### Recommendations for Improvement

1. **Diversify the Sample**: Conduct follow-up studies with a more representative sample, including older adults, non-students, and individuals with varying levels of technological proficiency. Discuss the limitations of the current sample in greater detail.

2. **Strengthen the Literature Review**: Include a more comprehensive review of AI in personal finance, focusing on decision-making support, trust, and cultural factors. Clearly articulate how the study builds on or diverges from prior work.

3. **Enhance Measurement Rigor**: Provide a detailed account of the questionnaire development and validation process. Justify the retention of low-loading items or consider removing them to improve construct validity. Streamline the questionnaire to reduce respondent burden.

4. **Address Methodological Gaps**: Discuss the implications of the long data collection period, report response rates, and include additional model fit indices. Address potential non-response bias and external influences on the data.

5. **Deepen the Discussion**: Engage more critically with the findings, particularly unexpected results (e.g., H4 rejection). Compare results with prior TAM studies and discuss implications for financial literacy, AI tool design, and policy in greater depth.

6. **Avoid Overgeneralization**: Frame the findings as context-specific (Poland, young respondents) and avoid broad policy recommendations without stronger evidence. Suggest cross-cultural or longitudinal studies to validate the results.

7. **Improve Clarity and Presentation**: Correct language errors, define acronyms, and ensure consistency in terminology. Provide detailed descriptions of figures and tables in the main text to enhance accessibility.

8. **Incorporate Socio-Demographic Analysis**: Analyze how age, gender, or education influence AI acceptance to provide a more nuanced understanding of adoption patterns.

#### Overall Assessment

The manuscript makes a valuable contribution by addressing the underexplored area of AI tools for consumer financial decision-making and applying an extended TAM framework. The methodology is robust, and the findings offer practical insights, particularly regarding the role of ease of use and financial literacy. However, the study’s impact is limited by the non-representative sample, superficial literature review, measurement concerns, and underdeveloped discussion. With significant revisions to address these weaknesses, the manuscript has the potential to be a strong addition to the literature on AI in finance. I recommend **major revisions** before reconsideration for publication.

Additional Notes

Ensure all supporting information (e.g., questionnaire, alternative translations) is clearly referenced and accessible in the final submission.

Verify the accuracy of URLs and DOIs in the reference list, as some (e.g., World Bank report) appear incomplete.

Consider discussing ethical considerations, such as privacy concerns with AI tools, given their relevance to consumer trust and adoption.

**Do you want your identity to be public for this peer review?** For information about this choice, including consent withdrawal, please see our Privacy Policy

Reviewer #1: **Yes:** Elahe Hosseini

Reviewer #2: No

---

## [Author Response · Author response to Decision Letter 1]

5 Jun 2025

The authors of the article would like to thank the reviewers for their insightful analysis and comments, which allowed us to significantly improve the quality of our article. We believe that after the changes we have made, reviewers will consider our article to be of appropriate scientific quality. We have carefully considered all the comments that were presented in the reviews and have addressed all of them.

---

## [Decision Letter · Decision Letter 1]

9 Jul 2025

Dear Dr. Buszko,

Thank you for submitting your manuscript to PLOS ONE. After careful consideration, we feel that it has merit but does not fully meet PLOS ONE’s publication criteria as it currently stands. Therefore, we invite you to submit a revised version of the manuscript that addresses the points raised during the review process.

**ACADEMIC EDITOR:**

Please address all the comments given by the reviewers.

We look forward to receiving your revised manuscript.

Kind regards,

Kashif Ali, PH.D

Academic Editor

PLOS ONE

Reviewers' comments:

Reviewer's Responses to Questions

**Comments to the Author**

Reviewer #1: All comments have been addressed

Reviewer #2: All comments have been addressed

2. Is the manuscript technically sound, and do the data support the conclusions?

Reviewer #1: Yes

Reviewer #2: Yes

3. Has the statistical analysis been performed appropriately and rigorously?

Reviewer #1: Yes

Reviewer #2: Yes

4. Have the authors made all data underlying the findings in their manuscript fully available?

Reviewer #1: Yes

Reviewer #2: Yes

5. Is the manuscript presented in an intelligible fashion and written in standard English?

Reviewer #1: Yes

Reviewer #2: Yes

Reviewer #1: Dear author(s),

Thank you for giving me the opportunity to review this paper. I agree that this is an important and pertinent topic. Although the idea is a good one, unfortunately, the way in which the study is operationalized holds back its potential contribution. There are a few areas where I would encourage the authors to give further thought, as follows:

INTRODUCTION

The introduction should clearly illustrate (1) what we know (the key theoretical perspectives and empirical findings) and what do we not know (major, unaddressed puzzle, controversy, or paradox does the study addresses, or why it needs to be addressed and why this matters). And, (2) what will we learn from the study and how does the study fundamentally change, challenge, or advance scholars’ understanding. Much sharper problematization is required so that the introduction draws the reader into the paper. The introduction therefore needs to do a better job in setting the stage for the articulation of the theoretical contributions of the study. At the end of the introduction, we should have a clear idea of what the paper is about (i.e. its motivation, the gap in understanding that the paper is trying to address and summary of theoretical contributions).With references of 2025- 2023-2024.

Paragraph 1, with no references, explaining the context of the research.

Paragraph 2, with references, explaining very generally what we know about the topic introduced in Paragraph 1.

Paragraph 3 explaining what we need to find out.

Paragraph 4 explaining briefly what this paper will do to find out, method etc.

Paragraph 5, with no references, explaining the structure of this paper.

The descriptions of the research philosophy and design adopted in the study need to be sufficiently developed. Research philosophy is a set of beliefs about collecting, analyzing, and using evidence concerning a phenomenon. Numerous research methods and philosophical frameworks are included under epistemology, which refers to what is known to be accurate, as opposed to doxology, which refers to what is thought to be true. Research design is the blueprint for data collection, measurement, and analysis. The research design is the approach adopted to combine the numerous components of the study consistently and logically, thereby ensuring that they will successfully solve the research topic. Elaborate further on the study’s data collection and analysis processes.

The Discussion lacks a critical synthesis and comparison of the primary data with the literature. The purpose of the discussion section is to interpret and describe the significance of your findings in relation to what was already known about the research problem being investigated and explain any new understanding or insights that emerged from your research. The discussion connects to the introduction through the research questions, hypotheses, and the literature you reviewed. The Discussion should include a critical synthesis and comparison of the data with the literature. The discussion clearly explains how your study advanced the reader’s understanding of the research problem from where you left them at the end of your review of prior research.

The Conclusion does not adequately discuss the theoretical and managerial implications of the study. Summarize your thoughts and convey the larger significance of your research. Identify and discuss how a gap in the literature has been addressed and demonstrate the importance of your ideas. Introduce possible new or expanded ways of thinking about the research problem. Also, state the ideas for future research in the conclusion. Make sure you create 3 subsections in the Conclusion: 1) Theoretical Implications, 2) Managerial or Policy Implications, and 3) Ideas for Future Research.

Thanks

Reviewer #2: 3. Areas for Improvement

Sample Representativeness and Generalizability:

The study relies on a convenience sample of primarily young respondents (89.2% aged 18–24) from three Polish universities, which limits the generalizability of findings (Page 20). While the authors acknowledge this as a pilot study, the discussion of sample bias could be more robust, particularly regarding how the sample’s characteristics (e.g., high digital literacy) may influence results.

The manuscript does not report response rates or non-response bias, which are critical for assessing the representativeness of even a convenience sample.

Recommendation: Expand the Limitations section (Page 78) to discuss how the sample’s homogeneity (young, digitally literate students) may skew perceptions of AI tool acceptance. Provide an estimate of response rates or explain why this was not feasible. Clarify how the planned representative sample study will address these limitations.

Depth of Literature Review:

The literature review (Pages 15–18) is comprehensive but could better engage with recent studies on AI in consumer finance beyond robo-advisors. For example, the discussion of general conversational AI models (e.g., ChatGPT, Claude) is brief and lacks integration with specific findings (e.g., Lopez-Lira & Tang, 2023).

The review does not sufficiently critique the limitations of prior studies, such as their focus on institutional perspectives or specific AI applications (e.g., chatbots, voicebots).

Recommendation: Strengthen the literature review by incorporating more recent studies on conversational AI in finance (e.g., Schlosky et al., 2024) and critically discussing the limitations of existing research to better position the study’s contribution.

Methodological Details:

The prolonged data collection period (February to October 2024) is explained as due to academic semesters and summer breaks (Page 20), but the manuscript does not discuss potential temporal effects (e.g., changes in AI adoption trends during this period).

The adaptation of the General Attitudes Towards Artificial Intelligence Scale (GAAIS) is mentioned (Page 14), but details on how it was tailored to the financial decision-making context are lacking.

The decision to retain items with factor loadings below 0.7 (Page 68) is justified, but the manuscript could clarify why these items were theoretically important to retain.

Recommendation: Provide a brief explanation of why no significant events affected AI use during the data collection period. Include a subsection in the Methods section detailing the adaptation process for the GAAIS scale. Elaborate on the theoretical relevance of retained items with low factor loadings.

Discussion and Theoretical Implications:

The Discussion section (Pages 74–77) effectively compares findings with prior studies (e.g., Belanche et al., 2019; Singh & Kumar, 2025) but could further explore the theoretical implications of the rejected H4 (PU → BI). This finding challenges the traditional TAM and warrants deeper analysis of why perceived usefulness does not directly influence behavioral intention in the AI context.

The discussion of mediation effects (H7, H8) is clear but could be expanded to compare with other TAM-based studies in finance (e.g., mobile banking, QR payments).

Recommendation: Strengthen the Discussion by elaborating on the theoretical significance of the full mediation (PU → ATT → BI) and its implications for TAM in AI contexts. Compare this finding with other TAM studies in finance to highlight its uniqueness.

Clarity and Writing Quality:

The manuscript is generally clear but contains minor grammatical errors and awkward phrasing (e.g., “in contrary” on Page 71 should be “in contrast”).

Figures 1–3 are referenced but not embedded in the provided document, making it difficult to evaluate their content and clarity.

The Conclusions section (Pages 77–79) is concise but could better articulate the study’s contributions to both theory (e.g., extending TAM) and practice (e.g., specific design recommendations for AI tools).

Recommendation: Conduct a thorough proofreading to address minor grammatical errors. Ensure all figures are embedded or accessible in the manuscript. Expand the Conclusions to explicitly highlight the study’s theoretical contributions and provide more specific practical recommendations (e.g., types of educational interventions to improve PEOU).

4. Specific Comments

Abstract (Page 9): The abstract is concise but could better emphasize the study’s novelty. Consider adding a sentence on how the findings challenge traditional TAM assumptions (e.g., rejection of H4).

Introduction (Pages 10–14): The introduction is well-structured but could be streamlined by reducing repetitive references to AI’s general capabilities. Clarify why Poland is a unique context for this study (e.g., digital adoption trends, financial literacy levels).

Literature Review (Pages 15–18): The hypotheses are well-supported, but the discussion of conversational AI tools (e.g., ChatGPT) could cite more recent studies to strengthen relevance.

Methods (Pages 19–25): The demographic breakdown (Page 20) is detailed, but a table summarizing respondent characteristics would improve readability.

Results (Pages 67–73): The rejection of H4 (PU → BI) is a key finding and should be highlighted more prominently in the Discussion as a contribution to TAM literature.

Discussion (Pages 74–77): The comparison to prior studies is adequate but could include more examples of AI tools (e.g., Finchat, TurboTax) to illustrate practical implications.

References (Pages 80–84): The reference list is comprehensive, but some DOIs are formatted inconsistently (e.g., “doi-1org-100bc2fxf0579.han3.uci.umk.pl”). Ensure all references follow PLOS ONE guidelines.

**Do you want your identity to be public for this peer review?** For information about this choice, including consent withdrawal, please see our Privacy Policy

Reviewer #1: No

Reviewer #2: No

---

## [Author Response · Author response to Decision Letter 2]

28 Aug 2025

Once again, we would like to thank the two anonymous reviewers for their thorough review of our article, pointing out shortcomings and offering suggestions and recommendations to improve its quality. We have taken all your comments into consideration and have tried to revise the article and make changes as closely as possible to your suggestions. All the modifications in the manuscript are indicated in yellow. We hope that this time we have met the requirements. Thank you again for your scientific support and patience.

Below, we present our response to each of the comments made by the two reviewers.

Response to REVIEWER 1

Introduction

Much sharper problematization is required so that the introduction draws the reader into the paper. The introduction needs to do a better job of setting the stage for the articulation of the theoretical contributions of the study. The descriptions of the research philosophy and design adopted in the study need to be sufficiently developed.

We have significantly restructured the introduction, following the recommendation to use 5 paragraphs. We distinguished subsections concerning what we know, what we don’t know, and why the topic is important, as well as what we learn from the study. We make the introduction more precise by reducing a bit of the general context.

The discussion

The Discussion lacks a critical synthesis and comparison of the primary data with the literature.

We have restructured the discussion to provide a clearer synthesis and interpretation of the key findings. Now, we are comparing our primary data more explicitly with previous literature, focusing on areas of convergence and divergence.

Additionally, we have expanded the scope of referenced literature to include more recent and thematically relevant studies. This allowed us to contextualize the rejection of hypothesis H4 more effectively and discuss its implications within the broader TAM research field and the specific domain of AI in personal finance.

The conclusion

The Conclusion does not adequately discuss the theoretical and managerial implications of the study. Identify and discuss how a gap in the literature has been addressed and demonstrate the importance of your ideas. Introduce possible new or expanded ways of thinking about the research problem. Also, state the ideas for future research in the conclusion.

We have restructured and expanded the conclusions section and clearly distinguished three subsections:

(1) Theoretical Implications,

(2) Managerial and Policy Implications, and

(3) Directions for Future Research.

Response to REVIEWER 2

Sample representativeness and generalizability

The manuscript does not report response rates or non-response bias, which are critical for assessing the representativeness of even a convenience sample. Expand the Limitations section (Page 78) to discuss how the sample’s homogeneity (young, digitally literate students) may skew perceptions of AI tool acceptance. Provide an estimate of response rates or explain why this was not feasible. Clarify how the planned representative sample study will address these limitations.

We have added a paragraph related to the sample homogeneity bias issue in the materials and methods section with the references, as well as expanded the description in the limitations section. We have also included consideration of how the representative sample may relate to the results obtained from the current sample.

Due to the specific goal of our investigation, which was validation of the research questionnaire, verification of the proper understanding of the questions, and analysis of the relations between the main constructs, we did not focus on the impact of socio-demographic features. Also, the simplified sampling technique, which we used, did not give us a basis for deep socio-demographic considerations. Therefore, we decided to keep the description of the socio-demographic aspects rather limited to maintain the proper structure of the text (focusing on the aspects of perception, attitudes, intentions, and actual use of AI)

Because we collected the data from the respondents through the research questionnaire, which was accessed by the QR code placed on the websites of the three university faculties that took part in the research, we cannot express the precise response rate. We could use the number of responses divided by the total number of students per faculty, but that does not necessarily show the proper rate (not all the students are browsing the faculty website, and there could be other respondents, such as post-graduate students, professionals, and graduates who can do so). Sending the research questionnaire by email would give us the possibility to calculate the response rate, but according to our experience, survey distribution by email is even less effective for collecting data (students usually do not respond to such emails).

Depth of literature review

The literature review (Pages 15–18) is comprehensive but could better engage with recent studies on AI in consumer finance beyond robo-advisors. For example, the discussion of general conversational AI models (e.g., ChatGPT, Claude) is brief and lacks integration with specific findings (e.g., Lopez-Lira & Tang, 2023).

The review does not sufficiently critique the limitations of prior studies, such as their focus on institutional perspectives or specific AI applications (e.g., chatbots, voicebots).

We significantly expanded the literature review section, inserting new references.

Methodological details

The prolonged data collection period (February to October 2024) is explained as due to academic semesters and summer breaks (Page 20), but the manuscript does not discuss potential temporal effects (e.g., changes in AI adoption trends during this period).

The adaptation of the General Attitudes Towards Artificial Intelligence Scale (GAAIS) is mentioned (Page 14), but details on how it was tailored to the financial decision-making context are lacking.

The decision to retain items with factor loadings below 0.7 (Page 68) is justified, but the manuscript could clarify why these items were theoretically important to retain.

We have added a detailed explanation of the process of the scale adaptation in the section Materials and Methods. We have clearly expanded the theoretical justification for retaining items with lower factor loadings according to the Reviewer’s recommendation.

In the revised manuscript, we have expanded our explanation to clarify why the four retained items—PU7, PU8, PU12, and ATT7—were preserved despite factor loadings slightly below 0.7. While their removal did not improve internal consistency, composite reliability (CR), average variance extracted (AVE), or reduce variance inflation factors (VIF), our decision was grounded not only in statistical evaluation but also in the unique theoretical value of these items.

Each of them captures important aspects of consumer perceptions regarding AI-based tools in financial decision-making that are not redundantly represented by other items in the scale. Specifically, they address perceived benefits such as risk reduction, improved management of savings, rapid access to financial data, and AI’s potential superiority in certain financial tasks. These dimensions are integral to the constructs of perceived usefulness and attitude and align directly with the conceptual framework and research gap addressed in our study. Removing them would have compromised the content validity and the conceptual breadth of the model.

Discussion and theoretical implications

The Discussion section (Pages 74–77) effectively compares findings with prior studies (e.g., Belanche et al., 2019; Singh & Kumar, 2025) but could further explore the theoretical implications of the rejected H4 (PU → BI). This finding challenges the traditional TAM and warrants deeper analysis of why perceived usefulness does not directly influence behavioral intention in the AI context.

The discussion of mediation effects (H7, H8) is clear but could be expanded to compare with other TAM-based studies in finance (e.g., mobile banking, QR payments).

We have restructured the section to provide a clearer synthesis and interpretation of the key findings. We are comparing more explicitly our primary data with previous literature - we have added over a dozen references from the years 2023–2025.

Clarity and writing quality

The manuscript is generally clear but contains minor grammatical errors and awkward phrasing (e.g., “in contrary” on Page 71 should be “in contrast”).

The grammatical errors were checked by the native speaker. We hope all the errors were removed.

Figures 1–3 are referenced but not embedded in the provided document, making it difficult to evaluate their content and clarity.

Unfortunately, we have no possibility to insert the figures into the main text of the manuscript. According to the PLoS One publishing rules, the figures must be added as separate files – what we do when uploading the manuscript and its revised versions.

The Conclusions section (Pages 77–79) is concise but could better articulate the study’s contributions to both theory (e.g., extending TAM) and practice (e.g., specific design recommendations for AI tools).

We have significantly expanded the conclusions sections and clearly distinguished its three parts:

(1) Theoretical Implications,

(2) Managerial and Policy Implications, and

(3) Directions for Future Research.

Specific comments

Abstract (Page 9): The abstract is concise but could better emphasize the study’s novelty. Consider adding a sentence on how the findings challenge traditional TAM assumptions (e.g., rejection of H4).

We have modified the abstract to better emphasize the study’s novelty and the findings.

Introduction (Pages 10–14): The introduction is well-structured but could be streamlined by reducing repetitive references to AI’s general capabilities. Clarify why Poland is a unique context for this study (e.g., digital adoption trends, financial literacy levels).

We have reduced the Introduction by cutting the part of the general context of the AI. Moreover, we have added a short comment about Poland, explaining the country’s potential for the implementation and development of innovations in finance.

Literature Review (Pages 15–18): The hypotheses are well-supported, but the discussion of conversational AI tools (e.g., ChatGPT) could cite more recent studies to strengthen relevance.

We have clearly developed the literature review section and added new references to strengthen relevance.

Methods (Pages 19–25): The demographic breakdown (Page 20) is detailed, but a table summarizing respondent characteristics would improve readability.

The table has been added.

Results (Pages 67–73): The rejection of H4 (PU → BI) is a key finding and should be highlighted more prominently in the Discussion as a contribution to TAM literature.

We have expanded the interpretation of the rejected H4 hypothesis, highlighting its theoretical significance within the TAM framework. The revised section discusses how the full mediation of perceived usefulness through attitude challenges conventional TAM assumptions and opens new theoretical perspectives for AI-related contexts. We restructured the Conclusions section, inserting more details of our contributions to theory, managerial policy implications, and directions for future research.

Discussion (Pages 74–77): The comparison to prior studies is adequate but could include more examples of AI tools (e.g., Finchat, TurboTax) to illustrate practical implications.

We have restructured the Discussion section to provide a clearer synthesis and interpretation of the key findings. We now more explicitly compare our primary data with previous literature. We have also added over a dozen references from the years 2023–2025. Additionally, we slightly repositioned the examples of consumer-facing AI applications (e.g., Finchat, TurboTax Assistant, Cleo, Emma), now using them as illustrative support for our empirical findings rather than as a standalone contextual note. This helps to strengthen the practical implications of the observed mediation effect by showing real-world relevance.

References (Pages 80–84): The reference list is comprehensive, but some DOIs are formatted inconsistently (e.g., “doi-1org-100bc2fxf0579.han3.uci.umk.pl”). Ensure all references follow PLOS ONE guidelines.

We have revised with care the references again and corrected when needed. The one “doi-1org-100bc2fxf0579.han3.uci.umk.pl” is exactly as it is provided by the Journal that published the article.

---

## [Decision Letter · Decision Letter 2]

17 Nov 2025

Thank you for submitting your manuscript to PLOS ONE. After careful consideration, we feel that it has merit but does not fully meet PLOS ONE’s publication criteria as it currently stands. Therefore, we invite you to submit a revised version of the manuscript that addresses the points raised during the review process.

We look forward to receiving your revised manuscript.

Kind regards,

Dr. Zhiyuan Yu

Academic Editor

PLOS ONE

Journal Requirements:

Reviewers' comments:

Reviewer's Responses to Questions

**Comments to the Author**

Reviewer #1: (No Response)

Reviewer #2: All comments have been addressed

2. Is the manuscript technically sound, and do the data support the conclusions?

Reviewer #1: Yes

Reviewer #2: Yes

3. Has the statistical analysis been performed appropriately and rigorously?

Reviewer #1: Yes

Reviewer #2: Yes

4. Have the authors made all data underlying the findings in their manuscript fully available?

Reviewer #1: Yes

Reviewer #2: Yes

5. Is the manuscript presented in an intelligible fashion and written in standard English?

Reviewer #1: Yes

Reviewer #2: Yes

Reviewer #1: Thank you for giving me the opportunity to review this paper.

The Discussion lacks a critical synthesis and comparison of the primary data with the literature. The purpose of the discussion section is to interpret and describe the significance of your findings in relation to what was already known about the research problem being investigated and explain any new understanding or insights that emerged from your research. The discussion connects to the introduction through the research questions, hypotheses, and the literature you reviewed. The Discussion should include a critical synthesis and comparison of the data with the literature. The discussion clearly explains how your study advanced the reader’s understanding of the research problem from where you left them at the end of your review of prior research.

The Conclusion does not adequately discuss the theoretical and managerial implications of the study. Summarize your thoughts and convey the larger significance of your research. Identify and discuss how a gap in the literature has been addressed and demonstrate the importance of your ideas. Introduce possible new or expanded ways of thinking about the research problem. Also, state the ideas for future research in the conclusion. Make sure you create 3 subsections in the Conclusion: 1) Theoretical Implications, 2) Managerial or Policy Implications, and 3) Ideas for Future Research.

The author should write the limitation of this study, comparison with previous studies, and suggestions for future studies in the conclusion.

Reviewer #2: Dear Authors,

I am writing to inform you that, based on a thorough review of the uploaded manuscript (PONE-D-25-08396R2), titled "Acceptance of AI-based tools in consumer financial decision-making: An application of the extended Technology Acceptance Model," the revisions made by the authors adequately address the concerns raised by the reviewers. The responses to both Reviewer 1 and Reviewer 2 demonstrate careful consideration and implementation of the suggested improvements, including restructuring the introduction, enhancing the literature review with recent references, expanding the discussion and conclusions sections, addressing sample limitations, and clarifying methodological details.

Key enhancements noted include:

A more focused and structured introduction that better articulates the research gap and contributions.

Expanded discussion with critical synthesis of primary data against existing literature, including new citations from 2023–2025.

Clear delineation of theoretical, managerial, and policy implications in the conclusions, along with directions for future research.

Justification for retaining certain items in the measurement scale and handling of potential biases.

These changes have strengthened the manuscript's clarity, theoretical depth, and overall quality, making it suitable for acceptance in PLOS ONE. I recommend proceeding with publication, subject to any final minor copyediting as needed.

Should there be any additional queries or requirements, please let me know.

Best regards

**Do you want your identity to be public for this peer review?** For information about this choice, including consent withdrawal, please see our Privacy Policy

Reviewer #1: **Yes:** Elahe Hosseini

Reviewer #2: No

---

## [Author Response · Author response to Decision Letter 3]

25 Nov 2025

Once again, we would like to thank the reviewers for their thorough review of our article, which highlighted shortcomings and provided valuable recommendations for its improvement.

We have taken all the comments into consideration and have attempted to revise the article, making changes according to all the indications. All improvements to the manuscript are highlighted in yellow. Moreover, we reviewed the whole text once again and introduced minor linguistic and stylistic adjustments. We hope that these improvements have met the postulated requirements.

Thank you again for your scientific support!

Below, we present our response to each of the comments

Response to REVIEWER 1

The discussion lacks a critical synthesis and comparison of the primary data with the literature.

We rewrote the discussion section almost from scratch, interpreting and describing the significance of our findings in relation to the existing literature and to the latest knowledge about our research problem. We have also explained new insights that emerged from our research (primarily the attitudes mediation channel).

The discussion should include a critical synthesis and comparison of the data with the literature.

We supported our discussion section, particularly in verifying hypotheses, with literature sources, and made a critical synthesis. Each hypothesis discussion is divided into clear subsections.

The conclusion does not adequately discuss the theoretical and managerial implications of the study.

We rewrote the conclusion section almost from scratch, summarizing the paper's key points and emphasizing the significance of our research. We also addressed the gap in literature coverage and underlined the importance of our ideas. Moreover, we have created clear subsections of the Conclusion part, including Theoretical Implications, Managerial Implications, Limitations, and Ideas for Future Research

Response to REVIEWER 2

Thank you very much for the positive feedback and acceptance of the paper!

Tomasz Szopiński, Michał Buszko, Małgorzata Porada-Rochoń

---

## [Decision Letter · Decision Letter 3]

26 Feb 2026

Acceptance of AI-based tools in consumer financial decision-making: An application of the extended Technology Acceptance Model

PONE-D-25-08396R3

Dear Dr. Buszko,

We’re pleased to inform you that your manuscript has been judged scientifically suitable for publication and will be formally accepted for publication once it meets all outstanding technical requirements.

Kind regards,

Fakhar Shahzad, Ph.D.

Academic Editor

PLOS One

Additional Editor Comments (optional):

Reviewers' comments:

Reviewer's Responses to Questions

**Comments to the Author**

Reviewer #1: All comments have been addressed

2. Is the manuscript technically sound, and do the data support the conclusions?

Reviewer #1: Yes

3. Has the statistical analysis been performed appropriately and rigorously?

Reviewer #1: Yes

4. Have the authors made all data underlying the findings in their manuscript fully available?

Reviewer #1: Yes

5. Is the manuscript presented in an intelligible fashion and written in standard English?

Reviewer #1: Yes

Reviewer #1: Thank you for giving me the opportunity to review this paper.The reforms have been well done and are acceptable.

**Do you want your identity to be public for this peer review?** For information about this choice, including consent withdrawal, please see our Privacy Policy

Reviewer #1: **Yes:** Elahe Hosseini

---

## [Editor Report · Acceptance letter]

PONE-D-25-08396R3

PLOS One

Dear Dr. Buszko,

I'm pleased to inform you that your manuscript has been deemed suitable for publication in PLOS One. Congratulations! Your manuscript is now being handed over to our production team.

Kind regards,

on behalf of

Dr. Fakhar Shahzad

Academic Editor

PLOS One